# The unique *Legionella longbeachae* capsule favors intracellular replication and immune evasion

**Silke Schmidt**[1,2], **Sonia Mondino**[1¤], **Laura Gomez-Valero**[1], **Pedro Escoll**[1], **Danielle P. A. Mascarenhas**[3], **Augusto Gonçalves**[3], **Pedro H. M. Camara**[3], **Francisco J. Garcia Rodriguez**[1], **Christophe Rusniok**[1], **Martin Sachse**[4], **Maryse Moya-Nilges**[4], **Thierry Fontaine**[5], **Dario S. Zamboni**[3], **Carmen Buchrieser**[1] *

**1** Institut Pasteur, Université Paris Cité, Biologie des Bactéries Intracellulaires, CNRS UMR 6047, Paris, France, **2** Sorbonne Université, Collège Doctoral, Paris, France, **3** Department of Cell Biology, Medical School of Ribeirão Preto, FMRP/USP, Ribeirão Preto, Brazil, **4** UTechS UBI, Centre de Ressources et Recherches Technologiques, Institut Pasteur, Paris, France, **5** Biologie et Pathogénicité fongiques, Institut Pasteur, Paris, France

¤ Current address: Laboratory of Molecular and Structural Microbiology, Institut Pasteur de Montevideo, Montevideo, Uruguay
* cbuch@pasteur.fr

**Data Availability Statement:** All relevant data are within the paper, are submitted to public databases and are its Supporting information files.

**Funding:** This work is supported by the French Government (grants ANR-10-LABX-62-IBEID to

## Abstract

*Legionella longbeachae* and *Legionella pneumophila* are the most common causative agents of Legionnaires' disease. While the clinical manifestations caused by both species are similar, species-specific differences exist in environmental niches, disease epidemiology, and genomic content. One such difference is the presence of a genomic locus predicted to encode a capsule. Here, we show that *L. longbeachae* indeed expresses a capsule in post-exponential growth phase as evidenced by electron microscopy analyses, and that capsule expression is abrogated when deleting a capsule transporter gene. Capsule purification and its analysis *via* HLPC revealed the presence of a highly anionic polysaccharide that is absent in the capsule mutant. The capsule is important for replication and virulence *in vivo* in a mouse model of infection and in the natural host *Acanthamoeba castellanii*. It has anti-phagocytic function when encountering innate immune cells such as human macrophages and it is involved in the low cytokine responses in mice and in human monocyte derived macrophages, thus dampening the innate immune response. Thus, the here characterized *L. longbeachae* capsule is a novel virulence factor, unique among the known *Legionella* species, which may aid *L. longbeachae* to survive in its specific niches and which partly confers *L. longbeachae* its unique infection characteristics.

## Author summary

*Legionella longbeachae* can cause a severe pneumonia, known as Legionnaires' disease. In Australia and New Zealand, *L. longbeachae* is the predominant species causing up to 50% of all infections due to *Legionella*. However, *L. longbeachae* virulence factors are nearly

C.B.) and the "Fondation pour la Recherche Médicale" (grant EQU201903007847 to C.B.). S.S. is a scholar in the Pasteur-Paris University (PPU) International Ph.D. program and received stipends from the Institut Pasteur and the "Fondation pour la Recherche Médicale" (FDT202204015116). We gratefully acknowledge the kind financial support of the Institut Pasteur (Paris) and the Région Ile-de-France (program DIM1Health to UTechs PBI). Work in the D.S.Z. laboratory is financed by The São Paulo Research Foundation (FAPESP grant 2019/11342-6). The funders had no role in study design, data collection and analysis, decision to publish, or preparation of the manuscript.

**Competing interests:** The authors have declared that no competing interests exist.

unknown. Here, we show that *L. longbeachae* expresses a capsule that is a major virulence factor of this pathogen as it is important for virulence *in vivo* in mice and in the environment in its natural host *Acanthamoeba castellanii*. It dampens the innate immune response in mice and human cells. Our study sheds light on an understudied environmental pathogen and identifies a new virulence feature of *Legionellae*.

## Introduction

*Legionella longbeachae* is a rod-shaped Gram-negative bacterium that can cause Legionnaires' disease, a severe form of pneumonia. Like other *Legionella* species, *L. longbeachae* is a facultative intracellular bacterium that causes the disease by inhalation of contaminated aerosols [1]. *Legionella* spp. are typically found in aquatic environments, either as free-living bacteria or in biofilm communities [2,3]. *Legionella longbeachae*, however, is predominantly isolated from potting soils and infections have been associated with gardening activities. Thus, gardening can constitute a risk factor for infection with *L. longbeachae* [4,5]. The clinical symptoms of Legionnaires' disease caused by *L. longbeachae* are similar to those caused by *L. pneumophila*, the most common causative agent of Legionnaires' disease [6]. Like *L. pneumophila*, *L. longbeachae* critically depends on a highly conserved type IVB Dot/Icm secretion system (T4SS) to establish an intracellular infection and to build a replication niche, the so-called *Legionella*-containing vacuole (LCV) [7–10]. Analyses of the genomic content of the *Legionella* genus genome showed that it is highly diverse with an astounding 18,000 predicted T4SS effector proteins [11]. For *L. longbeachae*, about 220 effectors were predicted, but only about 34% are shared with the *L. pneumophila* effector repertoire [12,13]. Furthermore, in-depth analyses of the *L. longbeachae* genome revealed unique sets of genes and extensive genomic recombination events that reflect both an adaptation to its soil environment as well as dynamic genetic exchange with other microorganisms [13–15].

Interestingly, common laboratory mouse strains are resistant to *L. pneumophila* replication, except A/J mice which allow replication of *L. pneumophila* due to Naip5 mutations. In contrast, *L. longbeachae* effectively replicates in the lungs, disseminates, and causes death in mouse strains such as A/J, BALBc, and C57BL/6 mice [16,17]. It was hypothesized that this is due to the lack of flagella in *L. longbeachae*, as the detection of flagella expressed by *L. pneumophila* leads to a rapid activation of the NLRC4 inflammasome and clearing of the infection in BALBc or C57BL/6 mice [18–20]. However, a flagellum-deficient *L. pneumophila* strain Paris is not lethal for C57BL/6 mice, and it induces a robust pro-inflammatory cytokine response *in vitro* [19,21]. In contrast, mice infected with wild type (WT) *L. longbeachae* die within 6 days after infection [21]. *L. longbeachae* spreads from the primary site of infection (the lungs) to the blood and spleens of the animals. In contrast to *L. pneumophila*, *L. longbeachae* only induces a low pro-inflammatory cytokine response *in vitro* [21]. A unique feature identified in the *L. longbeachae* genome is the presence of a gene cluster of 48 kb predicted to code for a capsule [13]. It comprises 33 genes that are annotated as glycosyltransferases, enzymes for the synthesis of nucleotide sugar precursors, and an ABC transporter, likely for the export of the capsule. The transporter was found to be homologous to the capsule transporter of *Neisseria meningitidis* [13]. We thus hypothesized that the capsule encoded in the *L. longbeachae* genome may be responsible for the enhanced virulence of *L. longbeachae* in mice as compared to *L. pneumophila* [21].

Many bacterial pathogens such as *Klebsiella pneumoniae* [22,23], *Streptococcus pneumoniae* [24,25], *N. meningitidis* [26–28], or *Acinetobacter baumannii* [29,30] encode capsules in

their genomes and the roles of these capsules in infection have been studied extensively. Capsules can protect bacteria from adverse environmental conditions, or antimicrobial agents such as antibiotics or antimicrobial peptides [31–35]. Likewise, in pathogenic bacteria, capsules can mimic surface polysaccharides of mammalian cells, thus avoiding recognition by the host immune system [36]. For example, the capsule of *E. coli* K1 contains polysialic acids that are anti-phagocytic and protect the bacteria from complement-mediated killing [37,38].

In this study, we visualized and characterized the *L. longbeachae* capsule and analyzed its functional role in infection. Comparing a knockout mutant in the capsule transporter and a WT strain, we reveal that the capsule plays a key role during infection in mammalian and protozoan hosts. Furthermore, we provide evidence that the capsule is responsible for the low cytokine response in infected macrophages and mice. Thus, we demonstrate that the capsule is a novel virulence feature of *L. longbeachae*, which is unique among the known *Legionella* species.

## Results

### The *L. longbeachae* capsule cluster is unique among *Legionella* species

We first analyzed the presence of the putative capsule cluster and orthologous genes among 58 different *Legionella* species (Fig 1). This analysis shows that *L. longbeachae* is the only *Legionella* species to encode the entire 48 kb capsule cluster. Two other species, *L. massiliensis* and *L. gormanii*, carry genes similar to the ABC-type transporter genes *ctrBCD* and *bexD*. However, these two species lack most of the glycosyltransferases encoded by *L. longbeachae*. We did not find similar genes in the genomes of *L. pneumophila*, except for an orthologous gene of *llo3163*, a hydroxyacid dehydrogenase similar to *serA*, indicating that the capsule cluster is specific for *L. longbeachae* (S1 Table).

To deepen these analyses, we selected twelve publicly available *L. longbeachae* strains, comprising nine strains from serogroups 1 (sg1) and three strains from 2 (sg2). All of them contain a similar capsule cluster whose genomic organization and genomic location are conserved (S1 Fig). Only a small region in the central part of the capsule cluster shows few differences among the glycosyltransferases and a duplication of *galE* in the NSW150 and B41211CHC genomes (Figs 1 and S1). All twelve strains encode genes for the predicted ABC type transporter *ctrBCD* and *bexD* (*ctrA*-like). Furthermore, they all encode glycosyltransferases and nucleotide sugar precursor genes belonging to the same groups of enzymes (S2 Table). This extremely high conservation in gene content, sequence similarity, and in the genomic location suggest that the acquisition of the capsule cluster dated back to a common ancestor of *L. longbeachae*.

### The *L. longbeachae* capsule cluster genes share homology to soil-dwelling bacteria

The evolutionary history of the capsule cluster in *L. longbeachae* is not known. Thus, we performed BLAST analysis on all its genes using *L. longbeachae* strain NSW150 (sg1) as the reference. S1 Table lists the best hits obtained by BLAST search. Across the cluster, we find homologous genes from β-proteobacteria often found in soils, and of γ- and δ-proteobacteria such as the soil-dwelling *Burkholderia* spp., *Geobacter* spp., or *Pseudomonas* spp., but also from *Nitrococcus mobilis* or *Alteromonas* spp., which are present in marine environments. Thus, the capsule cluster may have been acquired from soil-dwelling bacteria.

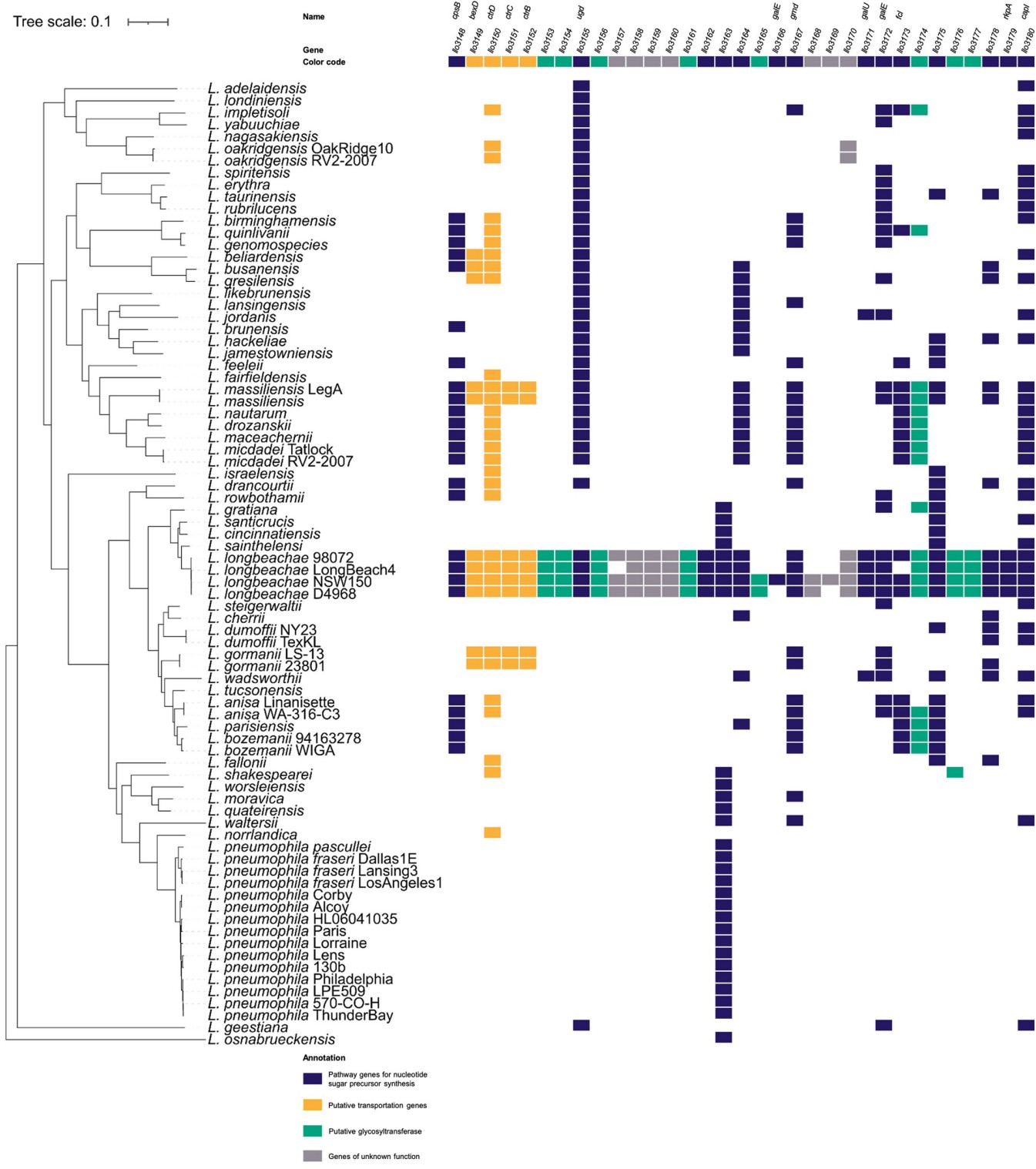

**Fig 1. Presence or absence of orthologous genes coding for the capsule in 80 *Legionella* species/strains.** Colored squares represent orthologous genes. Orthologous gene reconstruction was done with PanOCT: amino acid percentage identity cutoff was 30%, BLAST e-value cutoff $10^{-5}$, and minimum percentage match length of subject and query was 65%. For reference, the *L. longbeachae* NSW150 capsule cluster genes and gene names are shown on the top of the Fig. Color code: blue, pathway genes for nucleotide sugar precursor synthesis; yellow, putative transportation genes; green, putative glycosyltransferases; grey, genes of unknown function.

## Electron microcopy analyses confirm that *L. longbeachae* expresses a capsule that is absent from *L. pneumophila*

To further analyze the *L. longbeachae* capsule and to study its functional role, we constructed a mutant in the capsule transporter gene *ctrC* (*llo3151*) by replacing it with an apramycin cassette. We chose *ctrC* as a target, since a knockout mutant of a homologous gene in *Campylobacter jejuni* abrogated capsule expression [39]. When grown in ACES-buffered yeast extract broth (BYE) medium with or without apramycin, the Δ*ctrC* mutant did not display any significant growth differences as compared to the *L. longbeachae* WT (S2 Fig). To confirm that *L. longbeachae* indeed expresses a capsule and that the identified gene cluster is responsible for its expression, we used transmission electron microscopy (TEM). The *L. longbeachae* WT and the Δ*ctrC* mutant as well as *L. pneumophila* WT (negative control) were grown in BYE until the optical density $OD_{600}$ reached mid-log phase and late-log phase, referred to as exponential (E) phase and post-exponential (PE) phase, respectively. Cells were fixed and stained with cationized ferritin. TEM revealed a capsular layer surrounding the *L. longbeachae* WT (Fig 2A). The capsule was observed only in PE phase, indicating a growth phase dependent expression (S3A Fig). Its thickness was estimated to be between 50–100 nm, which is similar to ferritin-stained capsules observed in *E. coli* K30 [40]. In contrast, the Δ*ctrC* mutant strain was devoid of such a layer (Fig 2A) and *L. pneumophila* WT cells were not stained by the ferritin dye (Fig 2A). Complementation of the Δ*ctrC* mutant strain with a plasmid containing *ctrC* as well as the downstream genes *ctrD* and *bexD* under the control of their native promoter restored capsule expression (Fig 2B). We included the downstream genes as macrocolonies of a Δ*ctrC* mutant strain complemented with *ctrC* and *ctrD* only barely restored the WT phenotype compared to those complemented with the longer construct (S3B Fig). However, 50% of the complemented capsule mutant strains expressed a capsular structure indicating that regulation of capsule expression may be more complex (S3C Fig). We confirmed that there was no growth defect of the complemented mutant in liquid culture in comparison to the WT or Δ*ctrC* harboring the empty plasmid (S3D Fig). Taken together, our results confirm that, in contrast to *L. pneumophila*, *L. longbeachae* expresses a capsule and that *ctrC* is mainly responsible for its expression on the cell surface.

## The highly anionic *L. longbeachae* capsule is resistant to silver staining

To isolate the *L. longbeachae* capsule and to characterize its composition, we first used phenol-extraction for polysaccharide (PS) isolation followed by HPLC analyses. This approach allowed the identification of galactosamine, glucosamine, mannose and possibly quinovosamine in both the WT and the capsule mutant extracts (S4A Fig). After elution of the column, we detected high peaks in the WT sample that might correspond to phosphosugars. However, when visualizing these samples by silver or Alcian blue staining, we observed that phenol extraction seems to collapse the LPS structure, similarly to what has been seen for *L. pneumophila* [41]. Thus, *L. longbeachae* LPS neither nor CPS can be extracted by phenol (S4B Fig). Using an enzymatic isolation method [42] and silver staining revealed that *L. longbeachae* expresses a long O-antigen chain of different lengths between 30–70 kDa. In contrast, a different pattern was present for *L. pneumophila* with two shorter O-antigen fractions as shown previously, confirming that enzymatic PS isolation preserves the LPS structure in both *Legionella* species (Fig 2C) [43]. However, no differences between the WT and capsule mutant were observed. Therefore, we tested other dyes to visualize CPS. When using Stains-all dye, a carbocyanine dye that stains highly anionic compounds such as glucosaminoglycans [44], the enzymatically prepared *L. longbeachae* PS extracts showed a strong signal in the WT, but not the capsule mutant or *L. pneumophila* (Fig 2C). When we complemented

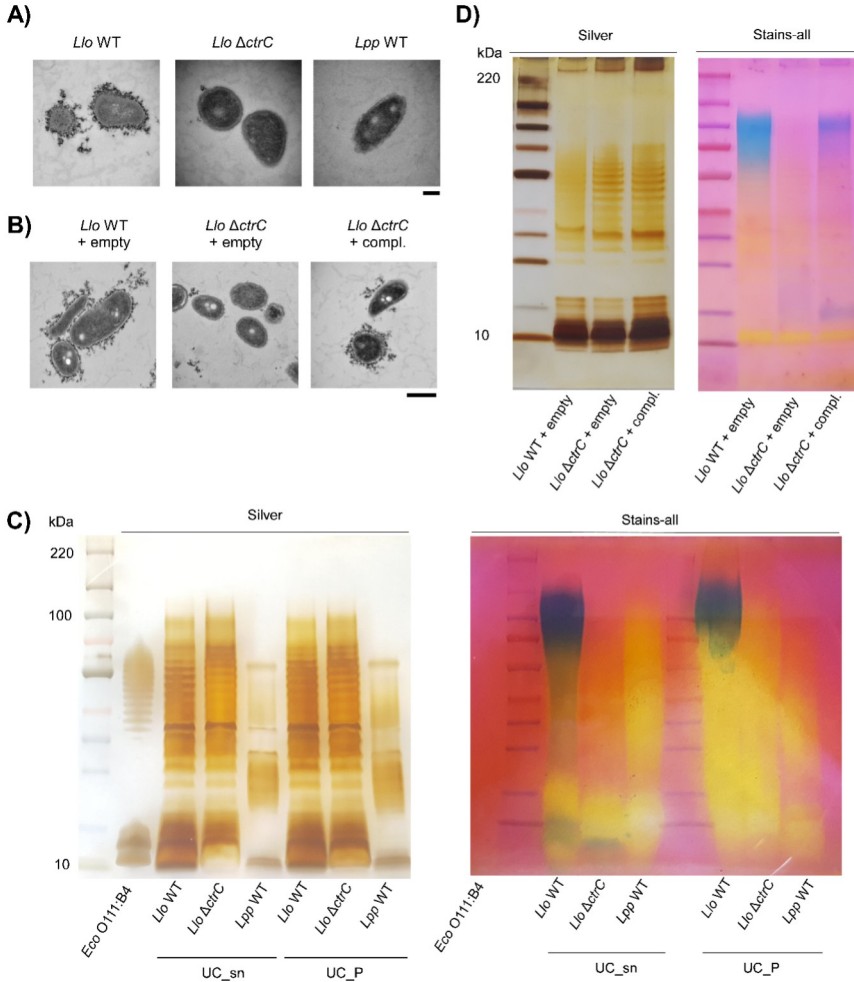

**Fig 2. The *L. longbeachae* capsule is expressed in exponential growth phase and contains a highly anionic polysaccharide that cannot be stained by silver staining. A)** *L. longbeachae* WT, Δ*ctrC* and *L. pneumophila* WT (negative control) were grown in BYE medium to post-exponential phase. Fixed cells were stained with cationized ferritin. TEM images are representative of n = 3 independent experiments. Scale bar = 200 nm. **B)** *L. longbeachae* WT or Δ*ctrC* harboring an empty plasmid (pBCKS) or the complementation plasmid (SSM083) were fixed in post-exponential phase and processed like samples in A). Representative images of n = 3 independent experiments. Scale bar = 500 nm. **C)** Polysaccharides were extracted from *Llo* WT, Δ*ctrC* or *Lpp* WT by enzymatic isolation and subjected to ultracentrifugation. SDS gels were stained with silver nitrate or Stains-all dye. Purified LPS from *E. coli* O111:B4 (Sigma) was used as a control for silver staining. **D)** Extracts from *Llo* WT or Δ*ctrC* harboring the empty plasmid (pBCKS) or the complemented mutant were stained by silver nitrate or Stains-all.

the capsule mutant, Stains-all reveals that the band observed in WT extracts is restored in the complemented mutant as well (Fig 2D). Again, silver staining did not detect any CPS in the complemented strain (Fig 2D). To test whether capsule expression may be temperature-dependent, we grew *Llo* WT, Δ*ctrC* and complemented Δ*ctrC* at 20°C in liquid medium. CPS extracted from bacteria grown at 20°C showed that also at 20°C the capsule is expressed in PE phase only in the *Llo* WT and complemented Δ*ctrC* (S4C Fig). The presence of a highly anionic PS corroborates our findings from staining with cationized ferritin for TEM and may explain our unique peak seen in HPLC. Taken together, the *L. longbeachae* capsule consists of highly anionic PS that cannot be stained by conventional silver staining methods.

### Encapsulated *L. longbeachae* is more sensitive to salt stress

The capsule may protect *L. longbeachae* from adverse environmental conditions. Thus, we grew the bacteria in liquid culture to PE phase and treated the cells with 0.05% Tween-20 with or without added NaCl or sucrose, as described previously [45]. After treatment with Tween-20, we did not observe a growth defect neither in the WT nor in the capsule mutant. However, in comparison to *L. pneumophila*, *L. longbeachae* seems inherently more resistant to detergent, independent of the capsule (S5A Fig). Similarly, both the *L. longbeachae* WT and the capsule mutant grew at similar rates after treatment with 2 mM or 10 mM $H_2O_2$. In comparison to *L. pneumophila*, the two *L. longbeachae* strains were more tolerant to high $H_2O_2$ present in the medium (S5B Fig). Thus, we conclude that *L. longbeachae* is inherently more resistant to detergent and oxidative stress than *L. pneumophila*.

It has been shown that salt stress in *L. pneumophila* is linked to virulence [46]. Indeed, a *L. pneumophila* mutant in the response regulator LqsR was shown to be less virulent but more resistant to salt stress than WT bacteria [46,47]. To test the role of the capsule under salt stress, we grew the *L. longbeachae* WT and capsule mutant as well as *L. pneumophila* WT to PE phase in liquid culture and plated the bacteria on BCYE in the presence or absence of 50 mM or 100 mM NaCl, respectively (S6 Fig). At high salt concentrations, both the capsule mutant and *L. pneumophila* exhibit a 3-log-fold growth difference. Surprisingly, however, there is also a 2-log-fold growth difference between the *L. longbeachae* WT, and the capsule mutant grown in the presence of 100 mM NaCl. Hence, similar to *L. pneumophila*, the analysis reveals that the less virulent *L. longbeachae* capsule mutant is more resistant to salt stress (S6 Fig). This growth difference was not detectable at a lower concentration of 50 mM NaCl, indicating that salt concentration is an important factor for optimal growth of both *Legionella* species.

### The capsule is expressed *in vitro* and during infection in a growth phase dependent manner

*Legionella pneumophila* is known to have distinct gene transcription profiles *in vitro* and during infection depending on the growth phase [48]. Thus, to learn when the genes encoding the capsule of *L. longbeachae* are expressed and if their expression is growth phase dependent, we performed RNAseq analyses. The *L. longbeachae* WT and the Δ*ctrC* mutant strain were grown in BYE until the $OD_{600}$ reached E or PE phase. Total RNA was isolated from the bacterial cells, depleted for rRNA, and sequenced to follow their transcriptional profile in both growth phases. This showed that almost all genes of the capsule cluster in the *L. longbeachae* WT strain were significantly downregulated in PE as compared to E phase (Fig 3A), indicating that the expression of the capsule on the cell surface is growth phase dependent. However, the capsule is visible only in the PE phase (S3A Fig) suggesting that the building of its complex polysaccharides and their export to the cell surface takes time and thus expression of the CPS (Fig 2) is highest in later growth phases. Moreover, when comparing *L. longbeachae* WT and the Δ*ctrC* mutant strain, apart from the expression of *ctrC* and the downstream genes of the capsule transporter operon, no significant differences in the transcription of the other capsule genes were observed neither in E phase (S7A Fig) nor in PE phase (S7B Fig, S3 Table). The transcription of the capsule in E phase and its expression in PE phase suggest that *L. longbeachae* synthesizes its capsule later in infection, probably to be prepared for cell attachment and a new infection of host cells, and/or for survival in the environment.

To test this hypothesis, we analyzed the capsule expression during infection, using a dual reporter to follow the transcription of the capsule transporter operon *in cellulo*. This dual reporter contains a constitutively expressed red fluorescent protein (mKate2) and two copies of superfolder GFP (sfGFP), each under the control of the native promoter of the capsule

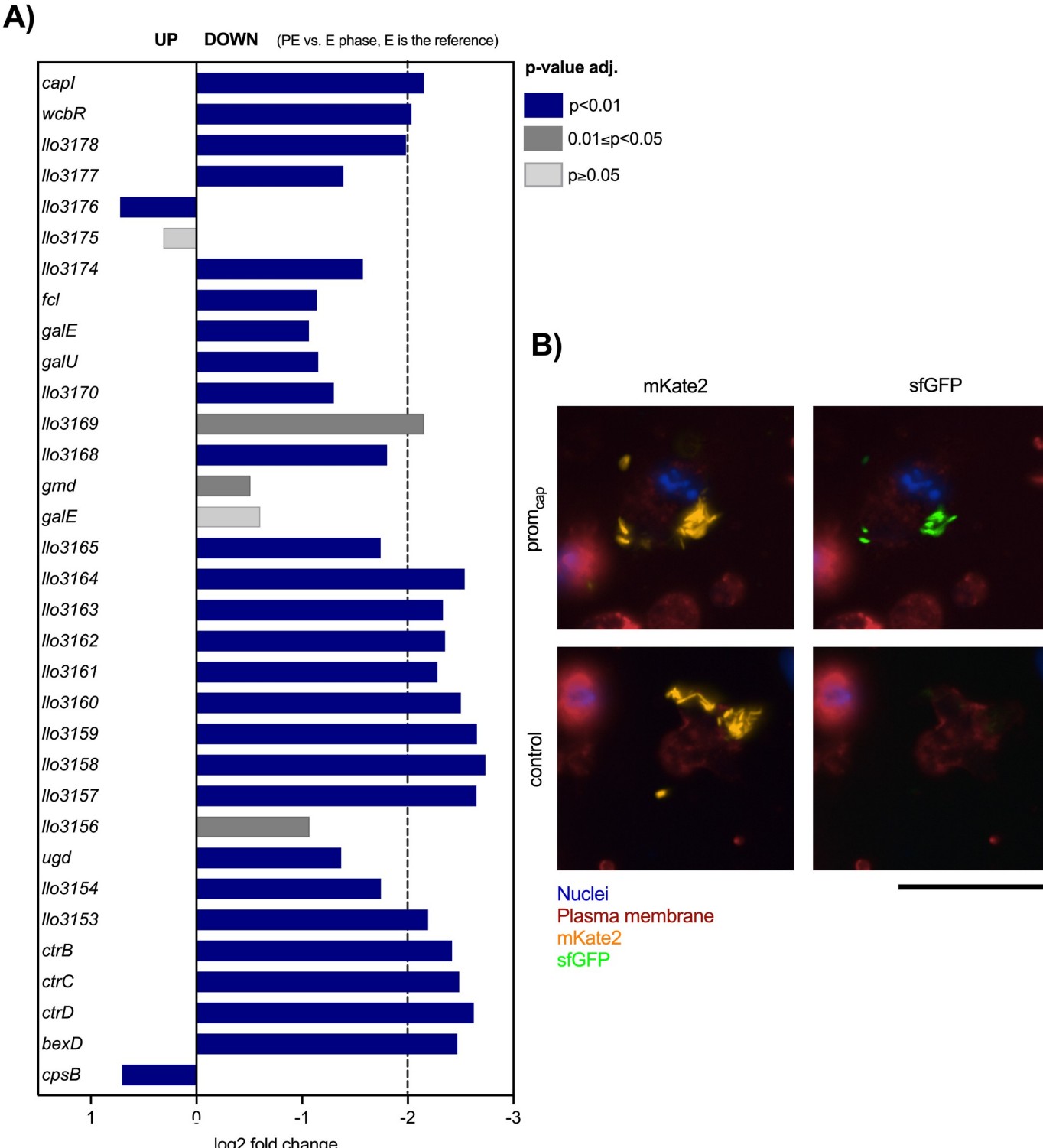

**Fig 3. The *L. longbeachae* capsule is transcribed in liquid medium and *in cellulo*, but it is not expressed in a Δ*ctrC* mutant. A)** RNAseq of WT transcripts (n = 4); E vs. PE (E is the reference); consider relevant genes with log2 fold change of ±2 and adjusted p value ≤ 0.05. **B)** THP-1 cells were infected with wild type *L. longbeachae* expressing constitutive mKate2 and inducible sfGFP under the control of the capsule promoter (prom$_{cap}$). A control plasmid containing constitutive mKate2 and two copies of sfGFP without the capsule promoter was included. Cells were imaged at 22 hours post infection. Scale bar = 50 μm.

transporter. As a negative control we used a plasmid containing constitutively expressed mKate2 and two copies of sfGFP without the native capsule promoter. These plasmids were transformed into *L. longbeachae* WT and used to infect THP-1 cells or *Acanthamoeba castellanii* (Figs 3B and S8, S1–S4 Videos). We followed the expression of sfGFP and mKate2 expression over time by live confocal imaging on an Opera Phenix system. We detected GFP expression starting at 12 hours that continuously increased in intracellularly replicating bacteria containing the cap$_{prom}$ promoter construct, but not the negative control. This revealed that the capsule is transcribed in both eukaryotic hosts. Thus, the capsule is universally transcribed *in vitro* and upon infection.

## The capsule is crucial for virulence *in vivo* in mice and in the environmental host *Acanthamoeba castellanii*

To learn if the *L. longbeachae* capsule plays a role in virulence *in vivo*, we infected mice with either the *L. longbeachae* WT or the Δ*ctrC* mutant at different bacterial loads and followed their survival over 10–15 days. Upon infection with the WT strain with $10^7$ bacteria, all animals succumb to the infection within 5 to 6 days, similar to what has been reported previously [21]. However, all mice infected with the Δ*ctrC* mutant survived the infection (Figs 4A and S9). To confirm that the observed phenotype is due to the missing capsule, we infected mice with the *L. longbeachae* Δ*ctrC* mutant either complemented or harboring an empty plasmid. All mice infected with the mutant strain carrying the empty plasmid survived the infection. In contrast, 40% of mice infected with the complemented strain died, indicating that complementation partially restored virulence *in vivo* (Fig 4B).

L. longbeachae* WT replicates to higher numbers in the murine lungs as compared to the capsule mutant, but both seem to reach the blood stream as we measured comparable levels of CFUs in the spleen (Fig 4C and 4D). Furthermore, we measured levels of lactate dehydrogenase (LDH) from bronchoalveolar lavage fluid (BALF) of infected mice to determine whether the capsule plays a role in lung damage. Indeed, the WT induces higher levels of LDH than the Δ*ctrC* strain (Fig 4E), indicating that the capsule is implicated in lung damage.

We then infected *A. castellanii* at a low MOI of 0.1 and at an environmental temperature of 20˚C with *L. longbeachae* WT, the Δ*ctrC* or the Δ*dotB* mutant strain. The latter is deficient in the T4SS and was used as a negative control. Replication of the bacteria was monitored by plating the bacteria every 24 hours over seven days. We observed a strong replication defect of the capsule mutant as compared to the WT, which becomes apparent at 72 hours post-infection (Fig 4F). As expected, the Δ*dotB* mutant failed to replicate in *A. castellanii* but seems to persist over extended periods of time indicated by the stable CFU counts towards the end of the experiment. Like in mouse infections, *A. castellanii* infected with the complemented strain restored replication of the bacteria to WT levels while the capsule mutant harboring the empty plasmid was impaired in replication (Fig 4G). We further tested the competitive fitness of the capsule mutant by performing a competition assay in *A. castellanii* as described previously [49]. This showed that the capsule mutant is completely outcompeted by the *L. longbeachae* WT strain between 24 to 48 hours of infection (Fig 4H), further underlining the importance of the capsule in virulence of *L. longbeachae* also in its environmental host.

An important question that arose from the *in vivo* infections was whether the strong virulence defect of the capsule mutant may be due to impaired effector secretion through the Dot/Icm T4SS in the Δ*ctrC* strain. We thus tested effector translocation using the beta-lactamase (BlaM) secretion assay as described previously [11]. We used a BlaM-fusion to the known T4SS effector RomA [50] and measured BlaM secretion in infected THP-1 cells by flow cytometry after 2 hours of infection. We included the *L. longbeachae* Δ*dotB* mutant as negative

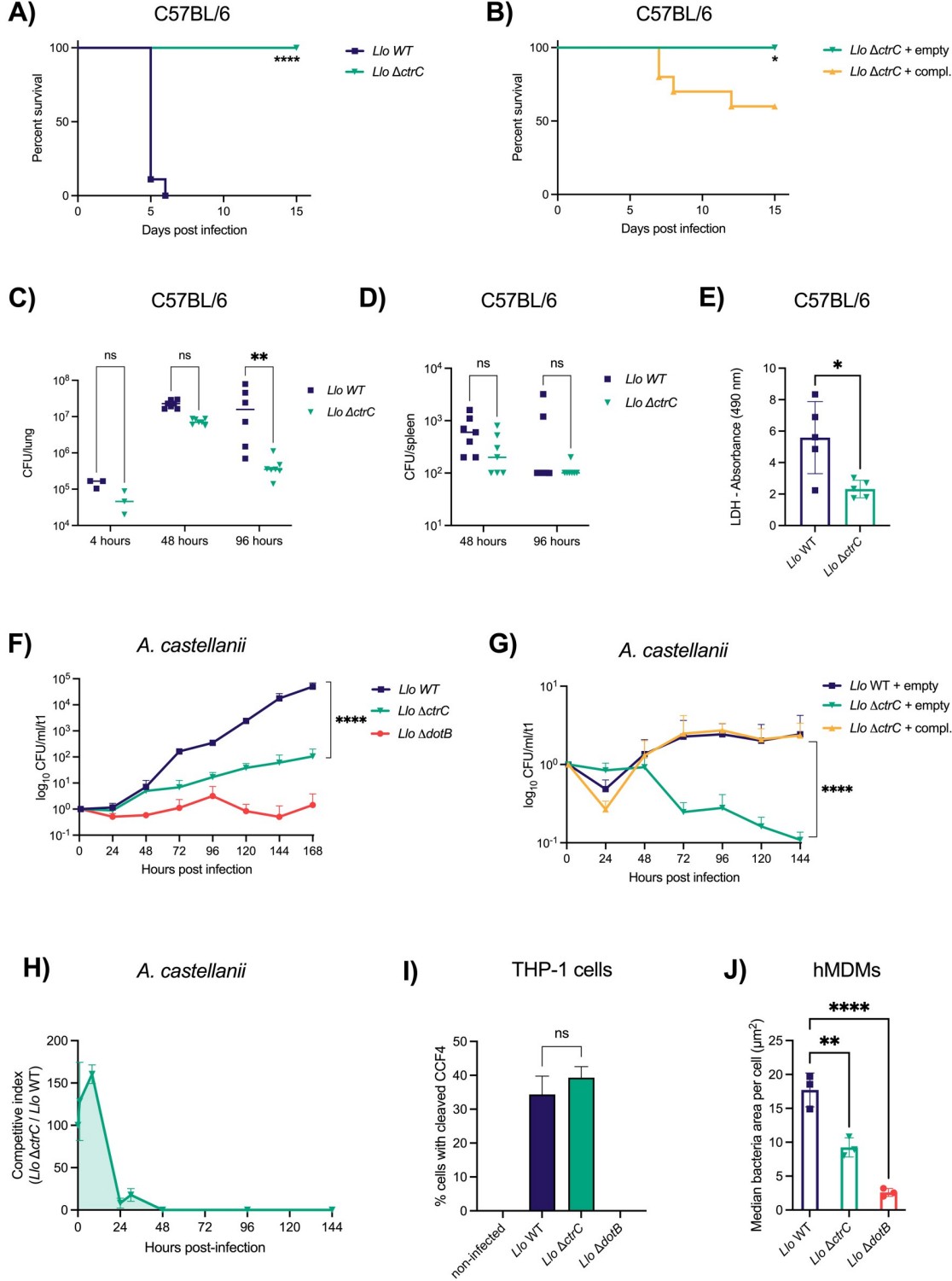

**Fig 4. The capsule mutant is impaired in replication and avirulent in mammalian and protozoan hosts, despite a functional Dot/Icm type IV secretion system. A)** Female C57BL/6 mice were infected with $10^7$ CFU of *Llo* WT or Δ*ctrC via* the nasal route. Survival was monitored over 15 days. Each group contained 9 mice. **B)** Infection of female C57BL/6 mice with the capsule mutant harboring an empty plasmid (pBCKS) or the complementation plasmid (SSM083) with $10^7$ CFUs. 10 mice per group. Statistical significance was determined by Log-rank (Mantel-Cox) test. *, p≤0.05; ****, p≤0.0001. **C)** CFUs from lungs of infected mice were plated at 4, 48, and 96 hours post-infection. Dots represent number of animals per group. **D)** CFUs from spleens of infected mice

were plated at 48 and 96 hours post-infection. Dots represent number of animals per group. Statistical analysis was performed by two-way ANOVA with Tukey's post-test. **E)** LDH activity test in converting L-lactate + $NAD^+$ to pyruvate + NADH, measured as absorbance at 490 nm. Lung lavage fluid from mice infected with *Llo* WT or Δ*ctrC* at 72 hours post-infection. Data points correspond to individual mice and are shown as means ± SD. Statistical significance was tested by unpaired t-test. ns, non-significant; *, p≤0.1; **, p≤0.01; **** p≤0.0001. **F)** *Acanthamoeba castellanii* trophozoites were infected at MOI 0.1 at 20˚C over 7 days. Samples were taken every 24 hours and CFUs are normalized to the input control. Data show means ± SD for n = 3 independent experiments. **G)** *A. castellanii* trophozoites were infected with *Llo* WT or Δ*ctrC* harboring an empty control plasmid (pBCKS) or the complementation plasmid (SSM083) at an MOI of 1. Samples were taken every 24 hours and CFUs normalized to t = 1. Data show means ± SD for n = 3 independent experiments. **H)** *A. castellanii* trophozoites were infected at MOI 0.1 with equal amounts of *Llo* WT or Δ*ctrC*. Samples were taken at 1, 8, 24, 30, 48, 96, 120, and 144 hours post-infection and normalized to the input control. Data show the competitive index of *Llo* Δ*ctrC* over *Llo* WT for n = 3 independent experiments. **I)** Translocation of the known T4SS effector RomA was tested in THP-1 cells infected with *Llo* WT, Δ*ctrC*, or a T4SS mutant, Δ*dotB*. Data represent means ± SD of n = 2 experiments. Statistical significance was tested by two-tailed t-test. **J)** Bacterial replication of *Llo* WT, Δ*ctrC* and Δ*dotB* in hMDMs at 20 hours post-infection. Data show median bacteria area ± SD of infected cells of n = 3 independent experiments. Statistical significance was tested by one-way ANOVA. ns, non-significant; *, p≤0.05, **, p≤0.01, ***, p≤0.001, ****, p≤0.0001.

control and *L. pneumophila* strain Paris as positive control. Both, the *L. longbeachae* WT and the Δ*ctrC* mutant translocated RomA successfully into the host cells to similar levels, whereas the Δ*dotB* mutant failed to translocate the effector (Fig 4I). Thus, the virulence phenotype of the capsule mutant *in vivo* and in *A. castellanii* is not due to an impaired Dot/Icm T4SS, as this strain maintains the ability to secrete effectors upon infection.

## The capsule provides a replicative advantage in primary human macrophages

To confirm that both *Llo* WT and Δ*ctrC* replicate inside the LCV, we infected U2OS cells constitutively expressing Sec61β-GFP, a marker of the ER. We followed the replication of the WT, Δ*ctrC* expressing dsRed and as negative control Δ*dotB*, a strain deficient in a functional T4SS, over time (S11 Fig). Both, WT and Δ*ctrC* replicate inside the cells building large vacuoles. More importantly, both strains recruit Sec61β early in infection, like *L. pneumophila* for which the recruitment of ER vesicles is a hallmark of intracellular replication.

Since *L. longbeachae* can cause the severe pneumonia Legionnaires' disease in humans, we also tested bacterial replication in human monocyte-derived macrophages (hMDMs). We observed that the WT replicates more efficiently in hMDMs than the capsule mutant as indicated by a larger, bacteria containing vacuole size in the WT infected cells (Fig 4J). To address the question whether the capsule plays a role in modulating cell death, we stained living, infected hMDMs for annexin V, an early marker of apoptosis. However, we did not observe significant difference in annexin V labelling of WT or Δ*ctrC* infected cells (S12A Fig). In addition, we tested whether the capsule may influence the induction of reactive oxygen species (ROS) in hMDMs. We labelled hMDMs with CellROX live cell dye to follow ROS production in infected cells, but we did not observe a significant difference in ROS production (S12B Fig). Interestingly, when we infected human macrophage-like THP-1 cells or murine bone marrow-derived macrophages *in vitro*, we observed no replication defect of the capsule mutant as compared to the WT, suggesting that successful infection of mice by *L. longbeachae* depends on the interaction of immune cells with the capsule *in vivo* (S13A and S13B Fig). These data show that there is a different replicative behavior in cell lines as compared to primary human or protozoan host cells as well as in the murine host.

Taken together, these results show that the capsule is important for replication in both protozoan and mammalian hosts and that the lack of the capsule renders the bacteria avirulent in mice.

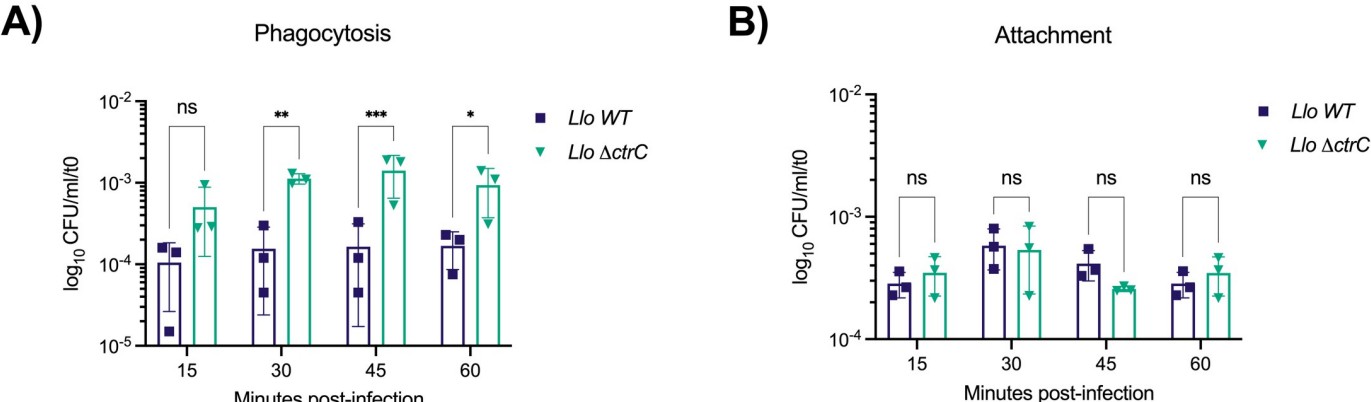

**Fig 5. The capsule delays phagocytosis of host cells. A)** For phagocytosis, differentiated THP-1 cells were infected with *Llo* WT or the *ctrC* mutant at MOI 10. Gentamycin was added at the indicated timepoints for 1 hour to kill extracellular bacteria and cells were lysed in water for CFU plating. Data show means ± SD of n = 3 independent experiments normalized to the input control at t = 0. **B)** For attachment, THP-1 cells were pre-treated with 2 μM cytochalasin D for two hours before infection with *Llo* WT or *ΔctrC* at MOI 10. At the indicated timepoints, gentamycin was added to half of the wells to kill extracellular bacteria. Cells were lysed in water and CFUs of gentamycin-treated cells (intracellular bacteria) and non-treated cells (total bacteria) were plated. Intracellular bacteria were deducted from total CFUs, normalized to the input control at t = 0 and data plotted as means ± SD of n = 3 independent experiments. Statistical significance was tested by two-way ANOVA with Tukey's post-test. ns, non-significant; *, $p \leq 0.05$, **, $p \leq 0.01$, ***, $p \leq 0.001$.

## The *L. longbeachae* capsule delays phagocytosis in human cells

Based on data published for other bacteria, we hypothesized that the capsule of *L. longbeachae* could be implicated in modulating phagocytosis or attachment to eukaryotic cells. To quantify phagocytosis, we incubated the bacteria with THP-1 cells and, after washing off unbound bacteria, we added gentamycin to kill extracellular bacteria and followed the infection by plating bacteria at different timepoints. For attachment studies, we pre-treated half of the cells with 2 μM cytochalasin D (cytoD), a known inhibitor of actin polymerization, and followed the infection by plating the bacteria recovered from the cytoD- as well as the non-treated cells. The overall CFU counts were normalized to the input control. We observed that WT bacteria were phagocytosed more slowly over time than the capsule mutant (Fig 5A). In contrast, both strains attached to THP-1 cells at similar rates, likely due to the synchronization of the infection by centrifugation at the start of each experiment (Fig 5B). This suggests that the capsule is important to delay phagocytosis by host cells.

Capsules have been shown to surround the bacterial cell wall and to mask LPS or outer membrane proteins like a shield. To further investigate if this could also be the case for *L. longbeacha*e, we tested agglutination by yeast mannan, a complex polysaccharide of the *Saccharomyces cerevisiae* cell wall that can bind bacterial fimbriae [51,52]. Indeed, the capsule mutant agglutinated, in contrast to the WT and the complemented mutant (S14 Fig). These results indicate that the capsule of *L. longbeachae* protects fimbriae and other attachment factors that are exposed on the surface of a capsule mutant.

## The *L. longbeachae* capsule impacts the pro-inflammatory cytokine response in primary innate immune cells

Capsules have been shown to mask cell structures like LPS or outer membrane proteins in bacteria. Due to their inherently low immunological recognition, capsules can have detrimental effects in infection settings such as reduced bacterial clearance or failure to mount an effective cytokine response against the pathogen [53]. Pro-inflammatory cytokines such as IL-6, TNFα,

or IL-1β can be induced upon engagement of pattern-recognition receptors such as Toll-like receptors (TLRs) or C-type lectin receptors (CLRs) by macrophages. To learn whether the *L. longbeachae* capsule plays a role in the cytokine response, we infected primary human mono-cyte-derived macrophages (hMDMs) with the *L. longbeachae* WT and the capsule mutant strains and measured cytokine levels in cell supernatants at 24 hours post-infection. Based on previous works [21], we expected low cytokine levels upon *L. longbeachae* infection. We there-fore opted for the high sensitivity SP-X array for human pro-inflammatory cytokines (Simoa). We observed a significant increase in IL-6 levels in the capsule mutant compared to the com-plemented strain (Fig 6A). Moreover, cytokine levels of the complemented strain were as low as those of the *L. longbeachae* WT strain. In contrast, we did not detect significant differences in TNFα or IL-1β levels (Fig 6B and 6C). These cytokines are known mediators of inflamma-tion in the context of *L. pneumophila* infections [54,55], and IL-6 and TNFα are highly induced upon infection with the *L. longbeachae* ΔdotB mutant. However, TNFα and IL-1β secretion in human macrophages does not seem to be driven by the capsule but are rather dependent on the presence of a functional Dot/Icm T4SS.

*L. longbeachae* is lethal for mice and fails to induce a strong cytokine response *in vitro* [21]. Our results show that the capsule plays a crucial role in virulence, and we therefore tested whether cytokine secretion during infection is also impacted by its presence or absence. We infected murine dendritic cells (DCs) with *L. longbeachae* WT or the capsule mutant. Similar to what was seen in human macrophages, the secretion of IL-6 was highly induced in the cap-sule mutant and lower in the *L. longbeachae* WT at 24 hours post-infection (Fig 6D). In con-trast to human macrophages, secretion of IL-1β was also induced in murine DCs infected with the capsule mutant as compared to the WT (Fig 6E). Using murine bone marrow derived mac-rophages (BMDMs), higher IL-6 and IL-1β were detected at 24 hours in cells infected with the capsule mutant compared to those infected with the WT or the complemented mutant (Fig 6G and 6H). However, we did not detect a significant difference in TNFα secretion in murine DCs or macrophages (Fig 6F and 6I). Thus, our results show that in both human and murine infection models the capsule dampens the secretion of pro-inflammatory cytokines.

When cytokine induction by the capsule mutant or the complemented mutant was tested in BMDMs, we observed that complementation of the capsule reduces TNF-α levels in BMDMs as compared to the capsule mutant, but it does not influence IL-6 levels *in vitro* (S15A and S15B Fig). Further, we tested cytokine levels in BALF from *Llo* WT or Δ*ctrC* infected mice. Here we observe higher IL-6 induction in the WT as compared to the mutant, but no differ-ence in TNF-α levels in BALF of infected mice at 72 hours post-infection (S15C and S15D Fig). However, we detected higher protein levels in BALF of WT infected mice as measured by Bradford assay (S15E Fig). These data indicate that proinflammatory cytokine induction *in vivo* is due to higher lung damage caused by the WT as shown by protein levels and lactate measurements.

## Discussion

In this study, we report that *L. longbeachae* expresses a capsule that is unique within the genus *Legionella* (Figs 1 and S1). Its genes seem to have been acquired from bacteria such as *Pseudo-monas*, *Klebsiella*, or *Neisseria* which are present in soil environments or the soil-dwelling bac-teria *Burkholderia* spp. and *Geobacter* spp., as evidenced by our comparative genomics analyses. This fits well with the environmental niche of *L. longbeachae*, as this bacterium is mostly isolated from moist soils and potting mixes [4,56,57]. Importantly, the genetic organi-zation of the cluster and its position in the genome are conserved among all *L. longbeachae* strains analyzed, suggesting that these genes have been acquired by horizontal gene transfer by

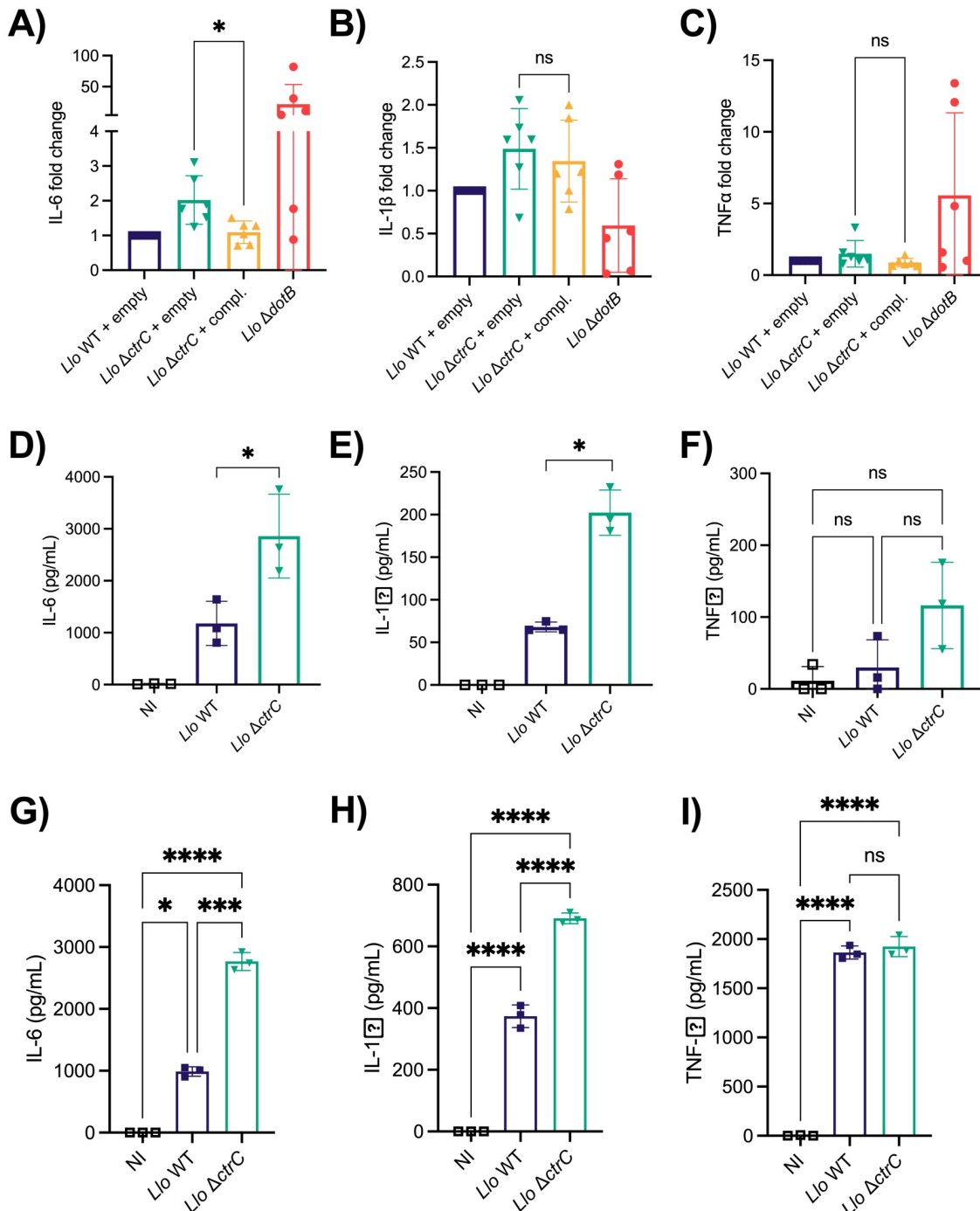

**Fig 6. The capsule influences the cytokine response of primary macrophages and dendritic cells. A-C)** Human monocyte derived macrophages (hMDMs) were infected at MOI 10 and cell supernatants were collected at 24 hours post-infection. High-sensitivity SP-X array was performed on cell supernatants. Data show means ± SD of fold change over *L. longbeachae* WT of n = 6 independent experiments. Statistical significance was tested by pairwise t-tests; ns, non-significant; *, p≤0.05. **D-F)** Murine bone-marrow derived DCs were infected at MOI 10 and cell supernatants collected at 24 hours post-infection. Data show means ± SD of n = 3 independent experiments. **G-I)** Murine bone-marrow derived macrophages were infected at MOI 10 and cell supernatants collected at 24 hours post-infection. Data show means ± SD of n = 3 independent experiments. Statistical significance was tested by one-way ANOVA with Tukey's post-test. ns, non-significant; *, p≤0.05; **, p≤0.01; ***, p≤0.001; ****, p≤0.0001.

a common ancestor of *L. longbeachae* (S1 Fig). However, the exact organism from which it has been acquired, probably *en bloc* given the high conservation in all strains, is not known as it has probably not been sequenced yet.

Transmission electron microscopy (TEM) confirmed that *L. longbeachae* expresses a capsule, which can be stained by cationized ferritin and that the *ctrC* mutant does not export the capsule (Fig 2A). CtrC is the inner membrane protein component in contact with CtrD, the nucleotide-binding domain containing ATPase of ABC capsule transporters. A deletion of the homologous *kpsM* gene in *Campylobacter jejuni* also abrogated capsule expression and impaired bacterial virulence in a ferret model of infection [39,58]. Interestingly, RNAseq analyses showed that the capsule cluster is highly transcribed in E phase, and TEM analyses revealed that the *L. longbeachae* capsule is visible in PE phase (S3 Fig). Thus, export of the capsule on the surface happens in later stages of bacterial growth, suggesting that the bacteria prepare themselves for a new infection and/or for survival in the environment. Indeed, using a fluorescence dual reporter to follow capsule transcription in living cells, we showed that the capsule transporter is expressed in both the human macrophage-like cell line THP-1 and in the environmental host *A. castellanii* upon infection (Figs 3B and S8).

Using conventional silver staining methods, we did not detect any fraction that corresponded to a possible CPS in the *L. longbeachae* WT, but the pattern looked highly similar to the LPS ladder in both the WT and in the capsule mutant strain (Fig 2C). Hence, depletion of the capsule does not impact bacterial LPS expression, which might have been an explanation for its low virulence (Fig 4). Only when we applied Stains-all dye, we detected a band at around 100 kDa in the WT, which is absent in the mutant and in *L. pneumophila* (Fig 2C). Stains-all can be used for differential staining of highly anionic compounds such as glycosaminoglycans like hyaluronic acid, chondroitin sulfate, dermatan sulfate, or heparin [44]. Possibly, we cannot detect the capsular polysaccharides by silver staining because it contains highly modified acidic sugars, which are not available for oxidation by periodate in silver staining. Indeed, it has been reported that some *C. jejuni* strains express hyaluronic acid-like and teichoic acid-like capsules, which often cannot be stained by silver, due to a high degree of O-methyl phosphoramidate groups in these CPSs [42,59,60]. Our results open new paths for further research into the biochemical composition of the *L. longbeachae* capsule, to determine what confers the highly anionic charge and its anti-phagocytic function.

Capsules have also been shown to mediate resistance to environmental stresses such as oxidative stress, detergent, or osmotic stress [32,61,62]. Interestingly, as shown here, *L. longbeachae* is inherently more resistant to oxidative stress and detergents than *L. pneumophila*, independent of the capsule (S5 Fig). These differences may be due to different LPS structures of these two *Legionella* species, which confer resistance to oxidative stress or detergents. During our attempts to isolate the CPS, we observed that *L. longbeachae* produces a longer O-antigen than *L. pneumophila* (Fig 2C), which may be an indication as to why this species is more resistant to $H_2O_2$ and Tween-20. In contrast, when we tested hyperosmotic stress due to NaCl, growth of the WT was highly impaired in the presence of 100 mM NaCl as compared to the capsule mutant and *L. pneumophila*, whose growth was less disturbed (S6 Fig). This is similar to what has been shown for the less virulent LsqR mutant of *L. pneumophila*, and our data point to a link between virulence of *L. longbeachae* and osmotic stress [47]. Capsules are hydrated shells surrounding bacteria, which may contain over 95% water [63] and which can protect them from hyperosmotic stress [61,64,65]. According to our TEM studies, the capsule is likely negatively charged. Thus, it is plausible that $Na^+$ ions present in high salt environments may disrupt the integrity of the capsule, causing membrane stress and leading to a growth defect in the WT.

Capsules have been shown to be important virulence factors of many Gram-negative bacteria that are human pathogens, such as *N. meningitidis* group B, *E. coli* K1 and K5, *K. pneumoniae*, or *Haemophilus influenzae* type b [66–71]. Using a mouse model of infection revealed that the capsule is a crucial virulence factor of *L. longbeachae* (Fig 4A). Remarkably, the capsule mutant was even completely avirulent in mice, a phenotype that was partially restored *in vivo* by complementation (Fig 4B). The virulence of *L. longbeachae* is linked to its capacity to replicate better in the lungs of infected mice and to cause higher lung damage as indicated by the release of LDH in lungs of infected mice (Fig 4C and 4E) and higher protein levels (S15E Fig). Previous studies have shown that *L. longbeachae* infection is lethal in common laboratory mouse strains [16,17,21]. Here, we provide evidence that the observed high virulence for mice is due to the capsule encoded by *L. longbeachae*. Thus, the *L. longbeachae* capsule is the first virulence factor described for human pathogenetic *Legionella* whose virulence phenotype is comparable with that of the loss of a functional Dot/Icm type IV secretion system.

Early studies on the pathogenicity of *L. pneumophila* revealed that the bacteria can replicate inside the amoeba and likely use the same mechanisms for infection of human cells [72]. Here, we show that the capsule mutant is impaired in intracellular replication in *A. castellanii*, and upon coinfection the WT rapidly outcompetes the capsule mutant (Fig 4F and 4H). Thus, the capsule of *L. longbeachae* is important for infection of both mammalian cells and environmental protozoa. Importantly, all *Legionella* species analyzed to date critically depend on the secretion of effector proteins into host cells *via* a highly conserved Dot/Icm type IV secretion system (T4SS), in order to establish infection and their replicative vacuole [7,10,73–76]. A *L. longbeachae* Δ*dotB* mutant having a nonfunctional T4SS cannot replicate in THP-1 cells, A549 cells and HEK cells *in vitro* [76]. Here, we show that it also fails to replicate in *A. castellanii* (Fig 4F). However, the capsule mutant is not impaired in effector translocation through the T4SS (Fig 4I). Thus, the observed infection phenotypes are not linked to a dysfunctional T4SS but to the lack of a capsule in the mutant strain.

Furthermore, we show that both WT and capsule mutant recruit Sec61β early in infection (S11 Fig). Sec61β is a marker of the ER and recruitment of ER-derived material to the LCV is a hallmark of *Legionella* infection and LCV formation [77]. Thus, both the WT and the capsule mutant replicate in the LCV. We further show that the WT replicates better in hMDMs as compared to the capsule mutant (Fig 4J). However, we did not observe any differences in apoptosis or ROS induction between the two strains (S12 Fig). This finding may be explained by the fact that both strains harbor a functional T4SS that allows the bacteria to secrete effector proteins into the host cell and to modulate host cellular pathways, such as apoptosis or ROS formation. Our data also point towards a difference in replication phenotypes depending on the infection models used. We observed a replication defect for the capsule mutant in primary cells such as hMDMs and amoeba as well as *in vivo*, but not in cell lines *in vitro* (S13 Fig). This result might point to the known metabolic differences between primary cells and cell lines, a topic that warrants further research.

Capsules have been shown to modulate phagocytosis by host cells by masking cellular receptors on the bacterial surface, or by impairing recognition of LPS by TLRs [78–80]. Indeed, in *L. longbeachae*, the capsule is involved in delaying bacterial phagocytosis by THP-1 cells (Fig 5). Moreover, when exposing bacteria to yeast mannan, which can bind proteins on bacterial cell walls leading to agglutination of bacteria [51,52], only the *L. longebachae* capsule mutant but not the WT or the complemented strain agglutinated (S14 Fig). Altogether, this shows that the *L. longbeachae* capsule is an anti-phagocytic factor and it shields bacterial outer membrane components from immune recognition. Further studies may shed light on whether the *L. longbeachae* capsule helps in evading the binding of complement factors, which may explain the difference in phagocytosis.

By avoiding immune recognition, bacterial capsules have been shown to dampen the host pro-inflammatory cytokine response [68,78,81]. The *Salmonella enterica* serotype Typhi Vi capsule dampens the expression of pro-inflammatory cytokines IL-6, TNFα, and IL8, and it prevents recognition by TLR4 [82,83]. A non-encapsulated strain of *K. pneumoniae* induced higher IL-6 levels in bronchoalveolar lavage fluid from infected mice than an encapsulated WT strain [84]. It was shown that the anti-stimulatory CPS1 capsule of the gut symbiont *Bacteroides thetaiotaomicron* dampens pro-inflammatory cytokines IL-6 and TNFα both in bone marrow-derived macrophages and dendritic cells, correlating with higher bacterial loads in the intestine [85]. Furthermore, it has been shown that the lack of capsule in *Campylobacter jejuni* leads to an increased release of IL-6 and TNFα in murine dendritic cells and macrophages [86,87]. Similarly, the *L. longbeachae* capsule modulates cytokine release in hMDMs, inducing a lower response of pro-inflammatory IL-6 in the WT as compared to the capsule mutant (Fig 6A). However, we did not detect any differences in secretion of TNFα or IL-1β in hMDMs. In contrast, when we infected murine dendritic cells and murine macrophages, the capsule mutant induced a higher immune response for IL-6 and IL-1β, but not TNFα (Fig 6D–6I), suggesting that the absence of capsule permits the liberation of bacterial-derived immunostimulatory components that would be only recognized by murine cells and not by human cells. Such immunostimulatory bacterial-derived components and their receptor counterparts in the host are yet to be identified, however an existing example of such differences between murine and human macrophages is the NAIP5 inflammasome, present in mice and absent in humans, which allows the exclusive recognition of flagellin from *L. pneumophila* by murine macrophages [88]. However, the leaked immunostimulatory component(s) cannot be flagellin as it is absent from *L. longbeachae*. Additionally, we observed that the complemented capsule mutant induces lower TNF-α than the capsule mutant harboring the empty plasmid (S15B Fig). *In vivo*, we see a reversed phenotype, the WT induces higher proinflammatory cytokine than the capsule mutant (S15C Fig). This is likely linked to the WT causing a high degree of lung damage (Figs 4E and S15E).

Thus, the *L. longbeachae* capsule dampens the immune response in innate immune cells *in vitro* by a mechanism that might involve the masking of immunostimulatory bacterial-derived components and their release, avoiding their recognition by innate immune pattern-recognition receptors.

In conclusion, our study provides evidence for a novel virulence mechanism of *L. longbeachae*, the expression of a capsule. This capsule is important for *L. longbeachae* replication in the environmental host *A. castellanii* and in mice. During infection, the *L. longbeachae* capsule modulates phagocytosis and dampens the innate immune response favoring bacterial replication. Importantly, our findings provide exciting new insights into how *Legionella* escape recognition by host cells and their defenses, and it opens new avenues to explore the architecture of this unique capsular type and to discover its anti-phagocytic molecules.

## Materials and methods

### Ethics statement

All mouse experiments were conducted according to the institutional ethical committees for animal care (Comissão de Ética em Experimentação Animal da Faculdade de Medicina de Ribeirão Preto FMRP/USP), approved protocol number 1248/2023.

### Bacterial strains, growth conditions, and cell culture

Bacterial strains used in this study are listed in Table 1. *Escherichia coli* DH5α subcloning efficiency bacteria were grown in Luria-Bertani broth or on LB agar. *Legionella longbeachae*

**Table 1. Bacteria and plasmids used in this study.**

| Bacteria | | |
|---|---|---|
| **Name** | **Source** | **Additional information** |
| *E. coli* DH5α | Invitrogen | Ref. 18265017 |
| *L. longbeachae* strain NSW150 | [13] | Wild type strain, serogroup 1 |
| *L. longbeachae* ΔctrC | This study | KO of *ctrC* in NSW150 background |
| *L. longbeachae* ΔdotB | [76] | Genomic deletion of *dotB* |
| *L. pneumophila* strain Paris | [104] | Wild type strain, serogroup 1 |
| **Plasmids** | | |
| **Name** | **Source** | **Additional information** |
| pGEM®-T easy vector | Promega | Cloning vector, Amp$^R$ |
| pBCKS+ | Stratagene | lacZ, Cm$^R$ |
| TOPO-mKate2 | Addgene | Ref. 68441, mKate2, Kan$^R$ |
| pLAW344 | [92] | Suicide plasmid incl. *sacB* cassette, Cm$^R$ |
| pSW001 | [105] | Constitutive dsRed, Cm$^R$ |
| pLGV012 | This study | pLAW344-*ctrC*::Apramycin |
| pXDC61 | [106] | N-terminal blaM, Cm$^R$ |
| pSSM012 | [50] | pXDC61-blaM-RomA |
| pSSM073 | This study | Complementation of *ctrC*, including *ctrD*, under the control of the native capsule promoter prom$_{cap}$, Cm$^R$ |
| pSSM083 | This study | Complementation of *ctrC*, including *ctrD* and *bexD*, native capsule promoter prom$_{cap}$, Cm$^R$ |
| pSS016 | This study | Dual reporter without prom$_{cap}$, Cm$^R$ |
| pSS017 | This study | Dual reporter incl. prom$_{cap}$, Cm$^R$ |

strains and *L. pneumophila* were cultured in N-(2-acetamido)-2-aminoethanesulfonic acid (ACES)-buffered yeast extract broth (BYE) or on ACES-buffered charcoal-yeast (BCYE) extract agar with antibiotics added where appropriate [89]. For the *L. longbeachae* ΔctrC mutant, 15 μg/ml apramycin (Sigma) were added to the medium. For plasmid maintenance, 5 μg/ml of chloramphenicol was added to the medium. For growth in minimal medium, cells were grown in CDM or MDM according to published protocols [90]. *Acanthamoeba castellanii* strain C3 (ATCC 50739) trophozoites were grown in PYG medium according to published protocols [13,91]. Infection buffer was prepared as PYG 712 medium [2% proteose peptone, 0.1% yeast extract, 0.1 M glucose, 4 mM MgSO$_4$, 0.4 M CaCl$_2$, 0.1% sodium citrate dihydrate, 0.05 mM Fe(NH$_4$)$_2$(SO$_4$)$_2$•6H$_2$O, 2.5 mM NaH$_2$PO$_3$, 2.5 mM K$_2$HPO$_3$] without proteose peptone, yeast extract, and glucose. The human monocyte cell THP-1 cell line (ATCC TIB-202) was maintained in RPMI GlutaMax supplemented with 10% fetal calf serum (FCS) at 37°C and 5% CO$_2$. For infections, undifferentiated THP-1 cells were seeded with RPMI and 50 μg/ml of Phorbol 12-myristate 13-acetate (PMA) to start differentiation into adherent cells. Cells were grown in the presence of PMA for three days and recovered overnight in fresh RPMI medium before infection.

## Construction of *L. longbeachae* ΔctrC

The 0.5 kb flanking regions of *L. longbeachae ctrC* (*llo3151*) were amplified using the primer pairs P3/P4 (upstream segment) and P1/P2 (downstream segment) using genomic DNA of *L. longbeachae* NSW150 as a template (see primers listed in Table 2). The apramycin cassette was amplified with primer pair Apra_s/Apra_as using pTOPO-ApraR plasmid as template. The three PCR fragments were ligated by PCR using primers P1/P3 and cloned into pGEM-T easy vector (Promega). Plasmid was then digested with NotI, and the combined *ctrC*::Apramycin

**Table 2. Primers used in this study.**

| Oligo name | Sequence | Direction |
|---|---|---|
| Apra_as | CCCTCCAACGTCATCTCGTTCTC | reverse |
| Apra_s | CATCAGCAAAAGGGGATGATAAGTTT | forward |
| P1 | TCCCGAGCTCAGTGAAGTCT | forward |
| P2 | AAACTTATCATCCCCTTTTGCTGATGACGCACCCATTACTCCATTC | reverse |
| P3 | TTGGGAAAACGCTCAGAAAC | forward |
| P4 | GAGAACGAGATGACGTTGGAGGGGGGCTCAAGAGCAAACCATA | reverse |
| SSM_79 | GGATCCCTCTCTCCCTTTACGGCGGTAT | forward |
| SSM_80 | TTGTTAGGAAATGCACATTTTGCATCGACACCAATCCTTAATGTCAAAA | reverse |
| SSM_82 | GGTACCTTAAGTTTGCTTGTTGTAAAATTCG | forward |
| SSM_85 | ATGCAAAATGTGCATTTCCTAAC | reverse |
| SSM_86 | GGCCGCTCTAGAACTAGTGGATCCCTCTCTCCCTTTACGGCG | forward |
| SSM_087 | ACTAAAGGGAACAAAAGCTGGGTACCTTAAGTTTGCTTGTTGTAAAATTCG | reverse |
| SSM_113 | GGAAGCTTACGAATTTTACAACAAGCAAACTTAAATAAGTTTTTACAGGAGTTAATTTTTAAAGTGATAAAG | forward |
| SSM_114 | ACTAAAGGGAACAAAAGCTGGGTACCCTAAGCCCTTATAACCTGTGTTGC | reverse |
| SSO_025 | GGCCGCTCTAGAACTAGTGGATCCACTCTCTCCCTTTACGGCG | forward |
| SSO_049 | CAGTTCTTCACCTTTACTCATCGACACCAATCCTTAATGTCAAAATTTC | reverse |
| SSO_050 | CTACAAACCAGGCATCAAATAGTAAAAGCTTACTCTCTCCCTTTACGGCG | forward |
| SSO_049 | CAGTTCTTCACCTTTACTCATCGACACCAATCCTTAATGTCAAAATTTC | reverse |
| SSO_051 | ATGAGTAAAGGTGAAGAACTGTTCAC | forward |
| SSO_052 | CGCCGTAAAGGGAGAGAGTAAGCTTTTACTATTTGATGCCTGGTTTGTAG | reverse |
| SSO_053 | GAACAAAAGCTGGGTACCTTACTATTTGATGCCTGGTTTGTAG | reverse |
| SSO_045 | TGGATGAACTCTACAAACCAGGCATCAAAGCTGCTAATGATGAAAATTATGCTGATGCT | forward |
| SSO_046 | GAGGTCGACGGTATCGATAAGCTTTTACTAAGAAGCATCAGCATAATTTTCATCATTAGC | reverse |
| SSO_047 | CATTTTTATCTATAATATTGGCAAATCTGCTACAGCGCTGCTAATGATGAAAATTATGCTG | forward |
| SSO_048 | CATTATTATTTATCCTGATTGATTCAGGTTATTTACTAAGAAGCATCAGCATAATTTTCATCA | reverse |
| SSO_065 | GCGGTGGCGGCCGCTTTACAGCTAGCTCAGTCCTAGGTATTATGCTAGCGAATTCGCTAGATTTAAGAAGGAGATATACATATGGTGAGCGAGCTGATTAAG | forward |
| SSO_066 | GGAGAGAGTGGATCCTCATCTGTGCCCCAGTTTGC | reverse |
| SSO_621 | GATGATGGATCCTGACTAACTAGCAGTAAAGGTGAAGAACTGTTCACC | forward |
| SSO_622 | GATGATAAGCTTTTAAGAAGCATCAGCATAATTTTCATC | reverse |
| SSO_623 | GATGATAAGCTTTGACTAACTAGCAGTAAAGGTGAAGAACTGTTCACC | forward |
| SSO_624 | GATGATGGTACCTTAAGAAGCATCAGCATAATTTTCATC | reverse |

fragment was ligated into the suicide plasmid pLAW344 [92] cut with the same restriction enzyme. The resulting plasmid, pLGV012, was introduced into *L. longbeachae* NSW150 as described above, and transformants were plated onto BCYE agar with apramycin. Resulting colonies were grown in BYE with apramycin and plated onto BCYE agar supplemented with apramycin and 5% sucrose. Sucrose-resistant/chloramphenicol-sensitive clones were screened, and successful deletion was verified through PCR and whole genome sequencing.

## Transformation of *L. longbeachae*

*L. longbeachae* was grown on BCYE agar plates at 37˚C. Bacteria from a fresh plate were washed three times with ice-cold 10% glycerol, and the final pellet was resuspended in ice-cold 10% glycerol. For transformation, a 400 µl aliquot of electrocompetent cells was freshly mixed with 300–600 ng of plasmid DNA and electroporated at 2.5 kV, 1000 Ω, and 25 µF. The

cultures were recovered in BYE broth at 37°C with shaking for 16 h and then plated onto BCYE agar with the appropriate antibiotic.

## Transmission electron microscopy

Bacterial strains were grown overnight in BYE and $OD_{600}$ was followed constantly to determine E phase ($OD_{600}$ 2.0–2.5) or PE phase ($OD_{600}$ 3.7–4.2). Samples of 10 ml were taken at E and PE phase and centrifuged for 15 minutes at 500 g to remove medium. The medium was discarded, and cells were immediately fixed overnight in 0.1 M cacodylate fixation buffer containing 2.5% glutaraldehyde. Subsequently, fixed cells were washed in 0.2 M cacodylate buffer and stained with 1 mg/ml cationized ferritin (Sigma, F7879-2ML) for 30 minutes at room temperature. After another wash, cells were immobilized in 4% agar before osmium fixation. Agar blocks were sequentially dehydrated in increasing volumes of ethanol and embedded in resin and thin sections were sliced on a Leica UC7 ultramicrotome to 70 μm thickness. Resin slices were mounted and imaged on a Tecnai BioTWIN 20–120 kV transmission electron microscope.

## RNA sequencing and analysis

Bacteria were grown at 37°C in BYE and $OD_{600}$ was measured to follow their growth. When bacteria reached exponential phase ($OD_{600}$ 2.0–2.5), 2x5 ml were pelleted and snap frozen in dry ice and ethanol. Bacterial morphology was checked under a microscope. Similarly, pellets were collected from bacteria grown to post-exponential phase ($OD_{600}$ 3.7–4.2) and stationary phase the following day. Bacterial pellets were resuspended in Qiazol (Qiagen) and RNA was isolated according to the miRNA Mini kit (Qiagen). The samples were Turbo DNase digested (Thermo Scientific) and rRNA was depleted using the RiboCop rRNA Depletion Kit for Gram-negative bacteria (Lexogen). Depleted RNA was metal-catalyzed heat-fragmented using the RNA Fragmentation kit (Ambion). RNA quantity was measured using a Qubit 2.0 (Invitrogen) and size distribution was confirmed between 100–200 nt by BioAnalyzer (Agilent Technologies). Fragmented RNA was subsequently processed according to the TruSeq mRNA sample preparation guide by Illumina. RNAseq was performed using Illumina NextSeq 550 multiplex sequencing (Illumina). For analyzing capsule gene expression, we used FASTQ files containing single end reads generated by Illumina sequencing. Sequencing reads were processed with Cutadapt software (version 1.15) to remove adapters. Trimming was performed with Sickle (version 1.33, https://github.com/najoshi/sickle) with a quality threshold (Phred Score) of 20. Reads shorter than 20 nucleotides were discarded. Clean reads were aligned to the *Legionella longbeachae* NSW150 sequence using Bowtie2 (version 2.3.4.3), and only uniquely mapped reads were kept for the read counts. We used Samtools package (https://github.com/samtools/samtools) to build indexed BAM files from the mapping results. To count the number of reads overlapping each genomic feature, we used featureCounts from the Subread package (version 1.6.3). Only primary alignments were counted. Differential analysis between the different conditions was performed with the R package Sartools (version 1.3.0) using DESeq2 methods. The "median" option was used to compute size factors.

## Construction of complementation and dual reporter plasmids

Capsule cluster promoter was amplified using primers SSM_79 and SSM_80, and the region containing *ctrC* (*llo3151*) and *ctrD* (*llo3150*) was amplified using primers SSM_82 and SSM_85. Both PCRs were performed using genomic DNA of *L. longbeachae* NSW150 as a template. Fragments were then ligated by PCR, with primers SSM_86 and SSM_87. The obtained amplicon was cloned into pBCKS (Stratagene) by restriction-free cloning [93], resulting in

plasmid SSM073. To construct plasmid SSM083, *bexD* (*llo3149*) was amplified using primers SSM_113 and SSM_114, and subcloned into SSM073 vector by restriction-free cloning [93]. For construction of the reporter plasmid, the capsule transporter promoter (prom$_{cap}$) was PCR amplified using primers SSO_025 and SSO_049 (prom$_{cap}$1) or SSO_050 and SSO_049 (prom$_{cap}$2) (listed in Table 2). Superfolder GFP (sfGFP, gift from David Bikard) was PCR amplified using primers SSO_051 and SSO_052 (sfGFP1) or SSO_053 and SSO_054 (sfGFP2). A C-terminal degradation tag was added to each sfGFP construct by restriction-free cloning [93] using primers SSO_045 and SSO_046 or SSO_047 and SSO_048. The GFP constructs were fused to the capsule transporter promoter by overlap PCR, one copy with restriction sites for BamHI and HindIII and the second copy with restriction sites HindIII and KpnI. Each copy was first ligated into pBCKS and inserts were confirmed by sequencing. The second copy was subsequently ligated into the first copy pBCKS vector using restriction sites HindIII and KpnI. One copy of mKate2 (Addgene #68441) was fused to a strong promoter from the Anderson collection (Table 2) by overlap PCR using primers SSO_065 and SSO_066 and ligated into the two copy pBCKS vector through restriction sites NotI and BamHI. A control plasmid was constructed by replacing the two promoter regions cap$_{prom}$ with a triple stop codon by restriction-free cloning using primers SSO_621/SSO_622 and SSO_623/SSO_624, respectively. Plasmid DNA was isolated using Nucleospin Plasmid kit (Macherey Nagel) and all plasmids were confirmed by sequencing.

## Imaging of dual reporter in THP-1 cells and *A. castellanii*

Bacteria expressing the dual reporter constructs (pSS016 or pSS017) were grown to PE phase in BYE medium with 5 μg/ml chloramphenicol. Differentiated THP-1 cells were infected at an MOI of 10 in a μClear 96-well plate (Greiner) for 1 hour. Cells were washed three times in PBS and supplemented with fresh RPMI medium. Subsequently, cells were stained with CellMask Deep Red Plasma Membrane Stain (Invitrogen, C10046) at 5 μg/ml for 20 minutes and washed once. Nuclei were stained with Hoechst dye (Invitrogen, H3570) at 1 μg/ml. Live imaging was performed at 40x magnification using the Opera Phenix confocal microscope (PerkinElmer). Images were obtained every hour and analyzed using the Harmony high-content analysis software (PerkinElmer). *A. castellanii* were infected in infection buffer at an MOI of 10 for 1 hour at 25°C. After one hour, cells were extensively washed in PBS to remove extracellular bacteria and maintained in infection buffer at 25°C. Cells were imaged at 40x magnification using the EVOS inverted digital microscope (Thermo Fisher).

## Animals and *in vivo* infections

Mice used in this study were bred and maintained in institutional animal facilities of University of São Paulo—School of Medicine of Ribeirão Preto/SP. All mice were at least 8 weeks old at the time of infection and were in a C57BL/6 (Jax 000664) genetic background. For the survival and CFU experiments, approximately 10 and 7 mice per group were used, as indicated in the Figs 4A–4E and S9. For *in vivo* experiments, the mice were anesthetized with ketamine and xylazine (300 mg/kg and 30 mg/kg, respectively) by intraperitoneal injection followed by intranasal inoculation with 40 μl of RPMI 1640 containing bacteria. For CFU determination, the lungs were harvested and homogenized in 5 ml of RPMI 1640 in a tissue homogenizer (Power Gen 125; Thermo Scientific) [94,95]. Lung homogenates were diluted in RPMI and plated on BCYE agar plates containing streptomycin for CFU determination as previously described. For survival determination, mice were observed once a day with the measurement of their weight [21]. The care of the mice followed the institutional guidelines on ethics in animal experiments.

## Bone marrow-derived dendritic cells and macrophages

Bone marrow-derived dendritic cells (BMDCs) and bone marrow-derived macrophages (BMDMs) were generated from C57BL/6 mice as previously described [96,97]. Mice were euthanized and bone marrow cells were obtained from femurs and tibias. BMDCs were harvested from femurs and differentiated with RPMI 1640 (Gibco, Thermo Fisher) containing 20% Fetal Bovine Serum (FBS, Gibco) and 20 ng/ml of recombinant GM-CSF (eBioscience), 2 mM L-glutamine (Sigma-Aldrich), 15 mM Hepes (Gibco) and 100 U/ml penicillin-streptomycin (Sigma-Aldrich) at 37°C with 5% $CO_2$ for 7 days. The non-adherent/loosely adherent fraction was harvested by collecting the culture supernatant and carefully washing the plate with PBS. After centrifugation of the total volume collected, BMDCs were resuspended in RPMI 1640 supplemented with 10% FBS and plated as indicated. BMDMs were harvested from femurs and differentiated with RPMI 1640 (Gibco, Thermo Fisher) containing 20% FBS and 30% L929-Cell Conditioned Medium (LCCM), 2 mM L-glutamine (Sigma-Aldrich), 15 mM Hepes (Gibco) and 100 U/ml penicillin-streptomycin (Sigma-Aldrich) at 37°C with 5% $CO_2$ for 7 days. Of note, in some experiments LCCM was replaced for 10% of a conditional medium from 3T3 cells stably expressing mouse MCSF. Cells were detached with cold PBS, resuspended in RPMI 1640 supplemented with 10% FBS and plated as indicated.

## Replication and competition assays in *Acanthamoeba castellanii*

Amoeba trophozoites grown at 20°C were washed in infection buffer, seeded at $1\times10^6$/ml and left to adhere for one hour prior to infection. Bacteria were resuspended in infection buffer to an MOI of 0.1 and left to infect for 1 hour. Dilutions of an aliquot of input bacteria was plated on BCYE to determine CFUs used for infection (t0). After one hour of infection, amoeba were washed three times in PBS to remove extracellular bacteria and resuspended in infection buffer. Samples of 500 µl were taken at this timepoint (t1), centrifuged at 14000 rpm for 3 minutes and vortexed for one minute to break up amoebae. Samples were subsequently taken every 24 hours for seven days. Experiments were carried out in triplicates and CFUs were counted after three to four days of growth on BCYE at 37°C. Competition assay was carried out as previously described [98]. Briefly, *A. castellanii* ($5 \times 10^6$ per flask) in infection buffer was infected at an MOI of 0.1 with a 1:1 mix of wild type *L. longbeachae* and Δ*ctrC* mutant bacteria. The infected amoebae were grown for seven days at 37°C. Every two days, a sample of lysed amoebae was diluted 1:100 and used to infect a fresh flask of amoebae (100 µl homogenate per flask). Dilutions were plated on BCYE agar plates containing apramycin or not, to determine CFUs.

## Infection of U2OS cells

Human osteosarcoma epithelial cells stably expressing Sec61β-GFP (U2OS-Sec61β-GFP) for labelling of the Endoplasmic Reticulum [99] were cultured in DMEM + GlutaMAX (Life Technologies), supplemented with 10% heat-inactivated FBS (Life Technologies). 20000 cells were seeded in DMEM in 96-well Greiner µClear plates and allowed to adhere before infection. Cells were infected with *L. longbeachae* at MOI 100 for two hours and extracellular bacteria were washed off using PBS. Cells were stained with Hoechst dye (Invitrogen, H3570) at 1 µg/ml to label nuclei. Live imaging was performed at 63x magnification at 37°C and 5% $CO_2$ using the Opera Phenix confocal microscope (PerkinElmer). Images were analyzed using the Harmony high-content analysis software (PerkinElmer).

### Replication assays in THP-1 cells

Infection assays of THP-1 cells (ATTC: TIB-202) were done as previously described [100]. Briefly, cells were seeded and differentiated into macrophage-like adherent cells in 12-well tissue culture trays (Falcon, BD lab ware) at a density of 2 x $10^5$ cells/well. Stationary phase *L. longbeachae* were resuspended in serum free medium and added to cells at an MOI of 10. After 2 h of incubation, cells were treated with 100 μg/ml gentamycin for 30 minutes to kill extracellular bacteria. Infected cells were then washed before incubation with serum-free medium. At 2, 24, 48, and 72 h, THP-1 cells were lysed with 0.1% Triton X-100. The infection efficiency of the different *L. longbeachae* strains was monitored by determining the number of CFUs after plating on BCYE agar.

### Bacterial replication in BMDMs

For CFU determination, macrophages were seeded at $2\times10^5$ cells/well in 24-well plates and cultivated in RPMI 1640 with 10% FBS. Cultures were infected at a multiplicity of infection (MOI) of 10, centrifuged for 5 minutes at 200 ×g at room temperature. After 1 hour of infection, BMDMs were washed twice with PBS, and 1 ml of medium was added to each well. For CFU determination, the cultures were lysed in sterile water, and the cell lysates were combined with the cell culture supernatant from the respective wells. Lysates plus supernatants from each well were diluted in water, plated on BCYE agar plates, and incubated for 4 days at 37˚C for CFU determination [94,95].

### Beta-lactamase translocation assay

Translocation of T4SS effectors was performed as previously described [101]. Plasmids pXDC61 or SSM012 were electroporated into *L. longbeachae* strains or *L. pneumophila* strain Paris as a positive control. Triplicate wells of THP-1 cells were seeded in 96- well plates at $10^5$ cells/well and differentiated into adherent cells with 50 μg/ml PMA for three days. Cells were recovered in RPMI without PMA for another night. Freshly transformed bacteria were induced by addition of 1 mM IPTG and THP-1 cells were infected at MOI 50 with stationary phase bacteria. Cells were subsequently centrifuged to synchronize the infection. At 1h30 after infection cell were incubated with CCF4-AM (Life Technologies) and 0.1 M probenecid at room temperature in the dark. After another 1.5 hours, cells were detached with non-enzymatic cell dissociation solution (Sigma). Flow cytometry was performed using a MACSQuant VYB system (Miltenyi Biotec), with excitation at 405 nm (violet) and emission collection with filters at 525/50 nm (Green) and 450/50 nm (Blue). Flow data were analyzed by FlowJo 10 software (LLC). Gates for green and blue fluorescence were set based on uninfected cells without any treatment. Western blots were performed on protein lysates from induced bacteria to confirm expression of blaM (see S10 Fig for gating strategy and Western blot).

### Isolation of human monocyte-derived macrophages (hMDMs)

Ficoll gradient centrifugation (Lympholyte, Cedarlane) was performed to isolate human peripheral blood mononuclear cells (PMBCs) from freshly extracted blood from healthy donors. To isolate CD14+ cells, anti-hCD14 magnetic beads were used and isolated cells were differentiated to hMDMs by addition of 50 ng/ml rhM-CSF (R&D Systems) to XVivo medium (Lonza). After 3 days, medium was exchanged, and cells were differentiated for another 3 days in fresh XVivo supplemented with rhM-CSF. All donors gave written consent under the agreement C- CPSL UNT–No. 15/EFS/023 between the Institut Pasteur and EFS (L'Établissement français du sang), in accordance with articles L1243-4 and R1243-61 of the French Public

Health Code and approved by the French Ministry of Science and Technology. Supply and handling of human blood cells followed official guidelines of the agreement between the Institut Pasteur, EFS, and the regulation of blood donation in France.

## Bacterial replication, early apoptosis, and ROS production in hMDMs

Human monocyte-derived macrophages were seeded in XVivo medium (Lonza) at 37˚C and 5% $CO_2$. Cells were stained with CellTracker Blue (CMF2HC, Invitrogen) and Hoechst dye (Invitrogen, H3570) for 1 hour pre-infection, before washing the cells in PBS. Cells were subsequently infected with *L. longbeachae* expressing dsRed (plasmid SW001, kind gift from Hubert Hilbi) at MOI 10 for 1 hour and extracellular bacteria were removed by washing with PBS. To monitor ROS production, cells were stained with CellROX DeepRed (C10422, Invitrogen) for 30 minutes and dye was washed off as indicated by the manufacturer. To monitor early apoptosis, cells were stained with Annexin V AlexaFluor 488 dye for living cells (A13201, Invitrogen). Live cell imaging was performed using the Opera Phenix confocal microscope (PerkinElmer) and bacterial replication, early apoptosis, and ROS production in single infected cells were analyzed using the Harmony high-content analysis software (PerkinElmer).

## Attachment and phagocytosis assays

THP-1 cells (ATTC: TIB-202) were seeded at $4x10^5$ cells/well in 12 well plates in RPMI-10% FCS and differentiated for three days with 50 nM PMA. After 96 hours of differentiation, RPMI without PMA was used to recover the cells overnight. For attachment assays, cells were treated with 2 µM cytochalasin D (Sigma) for two hours prior to infection. Bacterial strains were grown in BYE overnight and bacterial growth was monitored until the bacteria reached post-exponential growth phase ($OD_{600}$ 3.7–4.2). The bacteria were diluted to reach an MOI of 10 in RPMI with or without 2 µM cytochalasin D for attachment assays. At the indicated timepoints, cells were washed three times with PBS to remove non-adhered bacteria and half of the wells were treated with 100 µg/ml gentamycin for one hour to kill extracellular bacteria (control for internalized bacteria). Cells were lysed by addition of $ddH_2O$ for 15' at 37˚C. Dilutions of CFUs were plated for both the gentamycin-treated and untreated wells to determine total CFUs vs. adhered CFUs, respectively. For phagocytosis assays, cells were washed at the indicated timepoints and treated with 100 µg/ml gentamycin for one hour to kill extracellular bacteria. Cells were lysed in $ddH_2O$ for 15 minutes at 37˚C and dilutions were plated to determine CFUs. An aliquot of bacteria used for infections was plated for each strain to determine CFUs at t0.

## Detergent, oxidative, and osmotic stress assays

To test the resistance of the capsule mutant to salt stress, bacteria were grown in BYE to PE phase ($OD_{600}$ 3.7–4.2) and diluted to $2x10^9$/ml. Ten-fold dilutions were spotted in triplicates onto BCYE plates supplemented with 100 mM NaCl or plain BCYE plates and bacteria were grown at 37˚C for 6 days. Tween-20 was tested as previously described [45]. Briefly, bacteria were grown in BYE medium to post-exponential growth phase ($OD_{600}$ 3.7–4.2), washed in ultrapure water and adjusted to an $OD_{600}$ of 2.5. Subsequently, cells were treated with 0.05% Tween-20 ± 300 mM of either NaCl or sucrose for 30 minutes at 37˚C with moderate shaking. Ten-fold dilutions were spotted in duplicates on BCYE plates and bacteria were grown at 37˚C for 6 days. Similarly, for oxidative stress, post-exponential bacteria were washed once, adjusted to $OD_{600}$ of 2.5, and incubated in the presence of 2 mM or 10 mM $H_2O_2$ for 30 minutes at 37˚C with moderate shaking. Ten-fold dilutions were spotted onto BCYE agar plates in duplicates and bacteria were grown at 37˚C for 6 days.

## Agglutination assay

Bacteria were grown to post-exponential phase ($OD_{600}$ 3.7–4.2), washed in PBS adjusted to an $OD_{600}$ of 2.5. Subsequently, agglutination was tested by adding 1 mg/ml yeast mannan (Sigma) to the bacterial solutions and incubation at 37˚C for 15 minutes. Cells were subsequently observed at 100x magnification on an EVOS inverted digital microscope (Thermo Fisher).

## Polysaccharide extractions, gel electrophoresis and HPLC analysis

Bacteria were grown to post-exponential growth phase in BYE medium and washed in PBS. For phenol extraction, cells were treated with 45% hot phenol for 30 minutes according to previous protocols [102]. The aqueous phase was subsequently extracted and extensively dialyzed against water to remove residual phenol. To remove nucleic acids and proteins, extracts were treated with DNase, RNase, and proteinase K, each overnight. After dialysis, extracts were ultracentrifuged using a Beckman Coulter Optima ultracentrifuge at 100 000 rpm for two hours. Samples were freeze-dried and resuspended in ultrapure water for analysis by Dionex and SDS gel electrophoresis. For enzymatic extractions of polysaccharides, bacterial cell pellets were treated with mutanolysin (Sigma) and lysozyme (Sigma) according to published protocols [42]. Cell debris was removed by centrifugation and extracts were treated with DNase, RNase, and proteinase K, and subsequently dialyzed against water. Following ultracentrifugation, extracts were resuspended in ultrapure water.

SDS gel electrophoresis was performed using pre-cast mPAGE 4–20% bis-tris gels (Merck) and 20 μl of PS extracts were mixed in Laemmli buffer and boiled at 95˚C for 10 minutes. For silver staining, SDS gels were fixed in 45% ethanol solution containing 0.5% periodic acid (Sigma). After extensive washes in water, gels were submerged in 0.1% silver nitrate solution in water for 15 minutes. Gels were rinsed in water and developed in 3% sodium carbonate solution with added formaldehyde to fix the silver dye. Developer was neutralized by addition of citric acid (Sigma). For Alcian blue staining, gels were stained in 0.1% Alcian blue solution in 40% ethanol/5% acetic acid for 1 hour and destained in acetic acid solution [58]. For Stains-all staining, gels were fixed in 50% ethanol/10% acetic acid solution for 1 hour and extensively washed in water. A 0.1% Stains-all solution was prepared in water and gels were stained in the dark for 30 minutes. Gels were subsequently destained in water. For analysis of monosaccharides. monosaccharides were first released by acid hydrolysis (TFA 4 N, 4 hours at 100˚C or HCl 6 N, 6 hours at 100˚C). After vacuum drying of the hydrolysate, monosaccharides were identified and quantified by high performance anion exchange chromatography (HPAEC) with a pulsed electrochemical detector and an anion exchange column (CarboPAC PA-1, 4.6 x 250 mm, Dionex) using 18 mM NaOH as mobile phase at a flow rate of 1 mL/min; glucosamine and galactosamine were used as standards [103].

## ELISA of human pro-inflammatory cytokines

Differentiated hMDMs were seeded at $3x10^4$ cells per well in XVivo (Lonza) and infected with different bacterial strains at an MOI of 10. Cell supernatants were collected at 24 hours post-infection and centrifuged to remove cell debris and extracellular bacteria. The cleared cellular supernatants were rapidly frozen on dry ice and stored at -80˚C until cytokine measurements were performed. ELISA was performed using the SP-X CorPlex Human Cytokine Panel 1 (Quanterix), a sandwich-based multiplex ELISA with higher sensitivity and broad range of detection of 10 human pro-inflammatory cytokines. The SP-X array was performed from 6 independent experiments according to the manufacturer's instructions, including triplicate standard wells and duplicate wells for each sample. The data was analyzed using the proprietary SP-X analysis software.

### ELISA of murine pro-inflammatory cytokines

For ELISA experiments, BMDMs and BMDCs were seeded in 24-well plates ($5 \times 10^5$ cells/well). Infections were performed in RPMI 1640 supplemented with 10% FBS. After the indicate time points, the supernatants were collected for cytokine determination using ELISA kits according to the manufacturer's recommendations (R&D and BD Bioscience).

## Supporting information

**S1 Table. List of the highest hits of the *L. longbeachae* capsule cluster genes against the NCBI database (Trembl).**
(PDF)

**S2 Table. Capsule cluster (CPS) gene identitiy among *L. longbeachae* strains compared to NSW150.**
(PDF)

**S3 Table. List of genes up-or downregulated in exponential (E) compared to post-exponetial (PE) phase in the *L. longbeachae* WT strain compared to the capsule mutant strain.**
(PDF)

**S1 Fig. The capsule cluster is conserved among different *L. longbeachae* strains.** Visualization of the gene content of the cluster capsule in serogroup 1 and serogroup 2 *Legionella longbeachae* strains using GeneSpy [107]. Genes coding for the capsule plus two flanking genes on either side are represented. To obtain homogeneous and comparable annotations, all the genome sequences were reannotated using PROKKA. Based on the new annotation files, gene names and biochemical functions are used to infer families, and a color is attributed for each one.
(TIF)

**S2 Fig. The *L. longbeachae* WT and capsule mutant grow at similar rates in BYE medium.** *Llo* WT or $\Delta ctrC$ ± apramycin were grown in BYE medium at 37°C and $OD_{600}$ was followed using a BioTek Synergy Plate Reader 2. Data show means ± SD of n = 3 independent experiments.
(TIF)

**S3 Fig. Growth phase dependent expression of the capsule and quantification of capsule complementation. A)** *Llo* WT bacteria were grown to E phase ($OD_{600}$ 2.0–2.5) or PE phase ($OD_{600}$ 3.7–4.2), fixed and stained with cationized ferritin for TEM imaging. Scale bar = 1μm. **B)** Macrocolonies of *Llo* WT or $\Delta ctrC$ harboring an empty control plasmid (pBCKS) or the complementation plasmids (SSM073 or SSM083). Cells were grown to PE phase and 10 μl were spotted onto BCYE plates for 6 days. Colonies were imaged using a Leica M80 Stereo Microscope with top light and 1x magnification. **C)** Quantification of encapsulated bacteria from TEM images presented in Fig 2B. **D)** *Llo* WT with the empty vector and $\Delta ctrC$ with the empty vector or the complementation vector were grown in BYE medium at 37°C and $OD_{600}$ was followed using the Tecan Infinite M Nano plate reader. Data show means ± SD of n = 4 independent experiments.
(TIF)

**S4 Fig. Characterization and visualization of *L. longbeachae* cell wall polysaccharides. A)** Elution profiles of High-Performance Anion Exchange Chromatography. Top graph, standard reference hydrolyzed with TFA. Middle and bottom panel, bacterial phenol extracts hydrolyzed with HCl and TFA. Blue line, *Llo* WT; green line, *Llo* $\Delta ctrC$; black line, *Lpp* WT. **B)** SDS

gel electrophoresis of phenol extracts stained with silver nitrate or Alcian blue. **C)** Gel electrophoresis of extracts from *Llo* WT, Δ*ctrC* or *Lpp* WT grown at 20˚C in BYE to different optical densities and stained with Stains-all dye. QuiN, quinovosamine; GalN, galactosamine; GlcN, glucosamine; Gal, galactose; Glc, glucose; Man, mannose; Xyl, xylose; UC_sn, supernatants after ultracentrifugation; UC_P, pellets after ultracentrifugation; E, exponential; PE, post-exponential.
(TIF)

**S5 Fig. *L. longbeachae* is inherently more resistant to detergent and oxidative stress than *L. pneumophila*.** Bacteria were grown to PE phase ($OD_{600}$ 3.7–4.2) in BYE medium for each experiment. **A)** Treatment with Tween-20 ± 300 mM NaCl or 300 mM sucrose. **B)** Treatment with 2 mM or 10 mM $H_2O_2$. Representative images of n = 2 independent experiments.
(TIF)

**S6 Fig. Growth of *L. longbeachae* WT and capsule mutant strains under salt stress, compared to *L. pneumophila*.** Cells were grown to PE phase (OD600 3.7–4.2) in BYE medium and spotted onto BCYE plates ± NaCl. Representative images of n = 3 independent experiments.
(TIF)

**S7 Fig. RNAseq data of *L. longbeachae* WT vs. capsule mutant in E and PE phase (WT is the reference).** **A)** Comparison of E phase bacteria. **B)** Comparison of PE phase bacteria. WT vs. Δ*ctrC* (WT is the reference); consider relevant genes with log2 fold change of ±2 and adjusted p value ≤ 0.05. n = 4 independent experiments.
(TIF)

**S8 Fig. The *L. longbeachae* capsule is transcribed upon infection of *Acanthamoeba castellanii*.** *A. castellanii* was infected with *Llo* WT bacteria harboring the dual reporter plasmid (pSS017) at MOI 10 and 25˚C for 1 hour. Cells were imaged 48 hours post-infection using an EVOS inverted digital microscope. Scale bar = 50 μm.
(TIF)

**S9 Fig. Survival of mice infected with $10^6$ bacteria. A)** Female C57BL/6 mice were infected with $10^6$ bacteria and survival was monitored over ten days. Survival was monitored for 9 mice per group.
(TIF)

**S10 Fig. Gating strategy for beta-lactamase translocation assay and Western Blot confirming the expression of BlaM-constructs in *L. longbeachae*. A)** Gating strategy for flow cytometry. Cells were gated on THP-1 cells, and single cells were gated by FSC and SSC. Within the single cell population, gates for the signal showing the blue channel and the green channel were set based on non-infected stained cells. **B)** Original Western Blot showing the expression of BlaM-constructs with the known T4SS effectors LphD and RomA in *L. longbeachae*.
(TIF)

**S11 Fig. The *L. longbeachae* WT and capsule mutant replicate in the LCV and recruit the ER marker Sec61b to the vacuole. A)** U2OS cells constitutively expressing Sec61-GFP were infected with *Llo* WT, Δ*ctrC* or Δ*dotB* at MOI 100 and LCV formation was followed over time. Note that both *Llo* WT and Δ*ctrC* recruit Sec61b early in infection (white arrows), but not Δ*dotB* (see image inlets). Green: Sec61b-GFP; yellow: *L. longbeachae*; blue: Hoechst dye. Representative images of n = 3 independent experiments. Scale bar = 50 μm.
(TIF)

**S12 Fig. Annexin V and ROS production in infected hMDMs. A)** hMDMs were infected at MOI 10 with *Llo* WT, Δ*ctrC* or Δ*dotB* and labelled with Annexin V dye for early apoptosis induction at 20 hours post-infection. **B)** hMDMs were infected at MOI 10 with *Llo* WT, Δ*ctrC* or Δ*dotB* and labelled with CellROX dye for ROS production at 20 hours post-infection. Data show mean fluorescence intensity (MFI) ±SD of infected cells for n = 3 independent experiments. Statistical analysis was performed by one-way ANOVA. ns, non-significant.
(TIF)

**S13 Fig. The *L. longbeachae* WT and capsule mutant replicate to similar levels in THP-1 cells and murine BMDMs. A)** Differentiated THP-1 cells were infected at MOI 10 for 1 hour and treated with gentamycin to kill extracellular bacteria. CFUs were plated every 24 hours and normalized to the input control. Data show means ± SD of n = 6 independent experiments **B)** Bone marrow-derived macrophages (BMDMs) were infected at MOI 10 and CFUs plated every 24 hours normalized to the input control. Data show means ± SD of n = 3 independent experiments.
(TIF)

**S14 Fig. The capsule mutant agglutinates in the presence of yeast mannan.** Bacteria were grown to PE phase (OD$_{600}$ 3.7–4.2), washed, and treated with 1 mg/ml yeast mannan for 15 minutes. Scale bar = 10 μm.
(TIF)

**S15 Fig. Cytokine induction by capsule mutant and complemented mutant in murine cells and *in vivo*. A-B)** Murine bone-marrow derived macrophages (BMDMs) were infected with the capsule mutant or complemented strain at MOI 10 for 24 hours. **A)** IL-6 levels in cell supernatants. **B)** TNF-α levels in cell supernatants. **C-D)** Cytokine induction *in vivo*. Bronchoalveolar lavage fluid (BALF) of infected mice was collected 72 hours post-infection and levels of IL-6 and TNF-α were measured for *Llo* WT and Δ*ctrC*, or non-infected mice. **E)** Protein levels in BALF were measured using Bradford assay from infected mice at 72 hours post-infection. Data points represent individual mice and show means ± SD. Statistical significance was tested by one-way ANOVA. ns, non-significant; *, p≤0.1; ** p≤0.01; ***, p≤0.001.
(TIF)

**S1 Video. Time-lapse confocal imaging of THP-1 infected with *L. longbeachae* WT harboring the dual reporter plasmid for capsule transporter expression (as shown in Fig 3B).** Channels shown: blue, nuclei; green, inducible GFP expression of dual reporter plasmid; red, cell membrane. Infections were performed as described in Materials & Methods. Frames were acquired every hour from 2–23 hours post-infection.
(MP4)

**S2 Video. Time-lapse confocal imaging of THP-1 infected with *L. longbeachae* WT harboring the dual reporter plasmid for capsule transporter expression (as shown in Fig 3B).** Channels shown: blue, nuclei; yellow, constitutive mKate2 of dual reporter plasmid; red, cell membrane. Infections were performed as described in Materials & Methods. Frames were acquired every hour from 2–23 hours post-infection.
(MP4)

**S3 Video. Time-lapse confocal imaging of THP-1 infected with *L. longbeachae* WT harboring the triple-stop control plasmid of the dual reporter shown in S1 and S2 Videos (as shown in Fig 3B).** Channels shown: blue, nuclei; green, inducible GFP expression of control plasmid; red, cell membrane. Infections were performed as described in Materials & Methods.

Frames were acquired every hour from 2–23 hours post-infection.
(MP4)

**S4 Video. Time-lapse confocal imaging of THP-1 infected with *L. longbeachae* WT harboring the triple-stop control plasmid of the dual reporter shown in S1 and S2 Videos (as shown in Fig 3B).** Channels shown: blue, nuclei; yellow, constitutive mKate2 of control plasmid; red, cell membrane. Infections were performed as described in Materials & Methods. Frames were acquired every hour from 2–23 hours post-infection.
(MP4)

## Acknowledgments

We thank Hayley Newton for kindly gifting us the *Llo ΔdotB* mutant and David Bikard for sfGFP. Further we thank the Institut Pasteur UTechs CB platform for cytometry and biomarkers for assistance with SP-X ELISA measurements.

## Author Contributions

**Conceptualization:** Carmen Buchrieser.

**Data curation:** Dario S. Zamboni.

**Formal analysis:** Silke Schmidt, Sonia Mondino, Laura Gomez-Valero, Pedro Escoll, Danielle P. A. Mascarenhas, Augusto Gonçalves, Pedro H. M. Camara, Christophe Rusniok, Martin Sachse, Maryse Moya-Nilges, Thierry Fontaine, Dario S. Zamboni.

**Funding acquisition:** Dario S. Zamboni, Carmen Buchrieser.

**Investigation:** Silke Schmidt, Sonia Mondino, Laura Gomez-Valero, Pedro Escoll, Danielle P. A. Mascarenhas, Augusto Gonçalves, Pedro H. M. Camara, Christophe Rusniok, Martin Sachse, Maryse Moya-Nilges, Thierry Fontaine.

**Methodology:** Thierry Fontaine.

**Project administration:** Carmen Buchrieser.

**Resources:** Francisco J. Garcia Rodriguez.

**Supervision:** Sonia Mondino, Thierry Fontaine, Dario S. Zamboni, Carmen Buchrieser.

**Validation:** Silke Schmidt, Dario S. Zamboni, Carmen Buchrieser.

**Writing – original draft:** Silke Schmidt.

**Writing – review & editing:** Sonia Mondino, Carmen Buchrieser.

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
