## [Decision Letter · Decision Letter 0]

13 Feb 2024

Dear Dr. Buchrieser,

Thank you very much for submitting your manuscript "The unique Legionella longbeachae capsule favors intracellular replication and immune evasion" for consideration at PLOS Pathogens. As with all papers reviewed by the journal, your manuscript was reviewed by members of the editorial board and by several independent reviewers. In light of the reviews (below this email), we would like to invite the resubmission of a significantly-revised version that takes into account the reviewers' comments.

The reviewers appreciate the presented importance of a Llo capsule in bacterial virulence. However, they raised several concerns on presentation of the experimental data aimed to support the main conclusion that the capsule is crucial for bacterial replication in host organisms.

According to all the reviewers’ comments, the authors need to substantially edit the manuscript (text/ data presentation/ figure design/ data organization) in addition to the requested experiments before submitting the revised version of the manuscript.

We cannot make any decision about publication until we have seen the revised manuscript and your response to the reviewers' comments. Your revised manuscript is also likely to be sent to reviewers for further evaluation.

Sincerely,

Tomoko Kubori, Ph.D.

Academic Editor

PLOS Pathogens

Matthew Wolfgang

Section Editor

PLOS Pathogens

Michael Malim

Editor-in-Chief

PLOS Pathogens

orcid.org/0000-0002-7699-2064

The reviewers appreciate the presented importance of a Llo capsule in bacterial virulence. However, they raised several concerns on presentation of the experimental data aimed to support the main conclusion that the capsule is crucial for bacterial replication in host organisms.

According to all the reviewers’ comments, the authors need to substantially edit the manuscript (text/ data presentation/ figure design/ data organization) in addition to the requested experiments before submitting the revised version of the manuscript.

Reviewer's Responses to Questions

**Part I - Summary**

Reviewer #1: The manuscript by Schmidt et al. reports the characterization of Legionella longbeachae capsule as a new virulence factor that confers L. longbeachae its virulence in vivo in mice and its natural host Acanthamoeba castellani. The manuscript's content is interesting, but the structure of the text impairs the reading. Paragraphs should be rearranged (e.g. salt stress in the middle of the in vivo/in cellulo experiments; paragraphs with eukaryotic cells, then mice, then cytokines on cells again). Mice experiments are the strongest and should represent the last part of the results. The title “The unique Legionella longbeachae capsule favors intracellular replication and immune evasion” doesn’t seem to fully fit the manuscript as no strong evidence is presented (no experiment on LCV, microscopy imaging on replication/evasion, no difference on THP-1/BMDM replication in FigS8).

Reviewer #2: The paper convincingly demonstrates that in contrast to Legionella pneumophila (Lp), Legioenlla longbeachae (Llo) expresses a highly anionic capsule and that capsule formation can be disrupted by a mutation in the gene ctrC. The manuscript furthermore convincingly demonstrates that a ctrC mutant is less virulent in both amoebal and murine hosts. In case of mammalian cell infection, cell culture data implicates a role for capsule in phagocytosis avoidance and limitation of a proinflammatory cytokine response. Therefore, evasion of an immune response may underlie the virulence potential of the capsule in mice. However, whether these cell culture observations are linked to what happens during an in vivo infection was not further explored.

The MS provides evidence that a ctrC mutant has a growth defect within amoeba but not within human monocytic THP1 cells or murine bone marrow derived macrophages – these data thus suggest that loss of virulence in amoeba vs the mammalian host is mechanistically distinct. The MS does not provide any mechanistic model to account for the virulence defect in amoeba. The latter question warrants at least at some further discussion.

The data presentation/ figure design/ data organization could be substantially improved to render figures more reader-friendly and the content of the entire MS more readily digestible. Not all experimental designs include all the expected controls and/or meaningful timepoints. Despite these various shortcomings, this work constitutes an intriguing and important contribution to the Legionella literature.

Reviewer #3: This study characterises a polysaccharide capsule produced by Legionella longbeachae (Llo). Capsules are unprecedented across the Legionella genus and so the discovery of this genetic locus and the composition of the capsule is of interest. The authors identified that capsule expression is growth phase dependent and that the capule comprises highly anionic polysaccharide, although they did not define the carbohydrate composition. A capsule mutant (delta ctrC) showed defects in intracellular replication in Acanthamoeba and reduced virulence in a mouse infection model, which defines the capsule as a major virulence factor of Llo. This adds to knowledge of the role of capsules in bacterial virulence although this has been shown previously for different pathogens. I think the authors need to reevaluate some of the conclusions as outlined in my comments.

**Part II – Major Issues: Key Experiments Required for Acceptance**

Reviewer #1: The authors stated that the capsule is expressed in eukaryotic hosts, but there is no explanation/experiment that could explain the benefit of having this capsule. It would be interesting to design Llo WT and ΔctrC expressing GFP (or immunostaining) to decipher their replicative behavior: are they both in LCV? Does the capsule favor the survival? Are the bacteria egressing the cells by killing them? Is the capsule used for abiotic adherence?

Reviewer #2: There are couple of ‘straightforward’ experiments that would significantly strengthen the MS, as they would interconnect some of the observations made in this MS and provide more insights into the virulence mechanism of the Llo capsule. While desirable, I don’t consider these experiments essential for acceptance of the MS:

- data in Fig6 indicate that the Llo capsule reduces immune stimulation as assessed by IL-6 and IL-1beta secretion in murine DCs and BMDMs. Do these immunomodulatory activities observed in cell culture correlate with similar observations in vivo, possibly indicative of a causative link? I think the study would be significantly strengthened if cytokines were measured in BALF of Wt vs ctrC mutant infected animals

- it is intriguing that the ctrC mutant has a cell-autonomous growth defect in amoeba but not in the tested mammalian cells. Why would that be? And by what mechanism would the capsule provide Llo with a competitive advantage within amoeba ( but not mammalian cells). It would be helpful if the authors could provide some discussion on this topic (and maybe streamline the current – somewhat meandering – version of the MS Discussion

- Is the cytokine phenotype functionally linked to the block in phagocytosis? A simple cytoD experiment would address this question

Reviewer #3: Major comments

1. Fig 4E, F, how was statistical analysis and standard deviation done on n=2? Surely n=3 is the minimum for this calculation.

2. Fig. 4D, does the capsule mutant survive and persist in broth similar to wt over this time, ie. is the effect just due to the presence of amoebae?

3. Although the authors show that phagocytosis of delta CtrC is increased in THP-1 cells, the mutant shows no reduction in replication in these cells and BMDM. The mouse virulence effects would suggest otherwise. I’m not sure I understand the explanation for this in the sentence (line 241) “In contrast, when we infected human macrophage-like THP-1 cells or murine bone marrow-derived macrophages in vitro, we observed only a slight replication defect of the capsule mutant as compared to the WT, indicating that successful infection by L. longbeachae depends on the interaction of immune cells with the capsule in vivo” What is the evidence for this, what immune cells are implicated here, or is the capsule mutant simply more susceptible to complement mediated bacterial killing in vivo? This could at least be tested in vitro

4. In Fig 5 and 6, the authors move away from THP-1 cells to primary macrophages and murine bone-marrow derived dendritic cells. These experiments have not controlled for bacterial load. If the increased phagocytosis in THP-1 cells of the CtrC mutant is consistent in these cells, this would directly relate to the cytokine response. ie. increased bacterial load leads to increased production of cytokines. Hence the idea that the capsule inhibits cytokine production or contributes to a low cytokine level is not correct.

5. Related to the point above, have the authors tested cell death of macrophages upon infection with wt and the ctrC mutant, which could also explain differences in cytokine secretion

6. Fig. A-C, why was fold change used here and not pg/mL which is the usual way to express a cytokine response. Biological relevance is much easier to judge with the latter.

7. Line 333, I’m not sure I understand statement that “IL-6 and TNF are highly induced upon infection with the L. longbeachae ΔdotB mutant. However, TNF and IL-1b secretion in human macrophages does not seem to be driven by the capsule but are rather dependent on the presence of a functional Dot/Icm T4SS” when there is no difference statistically and huge variability in data points among the strains/samples (Fig6A-C). The only significant result shown was in IL-6 between the capsule mutant and complemented strain (not even between the wt and crtC mutant). I think these results have been overinterpreted.

**Part III – Minor Issues: Editorial and Data Presentation Modifications**

Reviewer #1: The capsule is known to protect bacteria from antibiotics. Have you tried to test the potential sensitivity of the capsulated/non-capsulated strains for several antibiotics?

The paragraph “Encapsulated L. longbeachae is more sensitive to salt stress” could be placed after the “The highly anionic L. longbeachae capsule is resistant to silver staining.” as it interferes with the reading of the in vivo/in cellulo section.

Line 39-40/53: The word “crucial” is overstated.

Line 49: Missing “.”

Line 64: Rephrase – Gardening activity is a risk factor.

Line 85: Ref after infection

Line 106: Wild type is written in line 84 and an abbreviation should be put here.

Line 130-132: Rephrase, the word “indicate” should be replaced by “suggest”.

Line 140-142: The conclusion seems too strong compared to the data, you could qualify it.

Line 166-168: “80% of the cells express the capsule…” instead.

Line 168-170: Rephrase the conclusion.

Line 230: “and” instead of “at”

Line 652: Typo

Line 1091: “images”

Line 1116: ΔctrC

Line 1119: “MOI XYZ”

Line 1134: ΔctrC or ctrC mutant

Correct “gentamycin/gentamicin” in the text and figure legends.

References should be fixed (ex ref 36, 41, 78)

Figure 1: The names of the genes are missing.

Figure 2: C) “100” on the ladder

Figure S2: ΔctrC

Figure S3: The legend doesn’t match the figure.

Figure S4: “Llo WT; red line” but it appears black on the figure.

Figure S8: Y-axis for the two figures is different. The y-axis of FigS7B is not expressed in log10CFU, is it on purpose?

Figure 2A/B: Have you tried to knock-out ctrB, ctrD or bexD to see if the capsule was missing and then do a complementation? The complementation only restores 50% of the phenotype, if you need two other genes to restore the capsule expression, your ctrC deletion might have impeded the expression of the genes downstream as well. These points should be discussed later in the manuscript as several other genes could be involved in the phenotype observed. “ctrC is mainly responsible for its expression on the cell surface” is then an overstatement at this stage.

Figure 2C/D: The capsule characterization should be strengthened. Are the exopolysaccharides surface-associated and/or secreted? Could the PS observed on WesternBlot? What are the genes involved in the first step of the synthesis of the polysaccharides?

Figure S2: Please add the strain ΔctrC+empty as it is used later in the manuscript.

Figure 3A/Figure S4: A control of the RNAseq could be included with RTqPCR on a few genes (for example ctrC, ctrD and bexD). As stated in the M&M, the exponential phase is reached at OD600 2.0-2.5 and the post-exponential phase OD600 3.7-4.2 but the stationary phase of the stains is obtained OD600 around 1.2 in FigS2. Is OD600 2.0-2.5 truly an exponential phase?

Figure S5: The mutant being ΔctrC, I would expect no data for the gene.

Line 204-206: The statement “This showed that almost all genes of the capsule cluster in the L. longbeachae WT strain were significantly downregulated in PE as compared to E phase (Figure 3A). This correlates with the visualization of the capsule in the PE phase (Figure S3A).” is challenging to apprehend. Is the expression of these genes that delayed compared to their transcription (timewise)? Overall, it would have been interesting to have an RNAseq on the intracellular bacteria at different time points to strengthen the statement.

Figure 3B: The authors should provide a quantification of the mKate2/GFP of the bacteria in several infected cells.

- Some bacteria seem to express mKate2 but not sfGFP, are those extracellular or intracellular?

- Does an antibiotic was used to kill extracellular bacteria for this experiment (not written in the M&M)? If gentamicin is used to kill extracellular bacteria; the MIC of each strain should be assessed.

- Is the GFP detected at 14 hours for the THP-1 and the amoebae? Does the GFP signal change over time? The authors are showing 22hrs and 24hrs images in Figures 3B and S6 but the GFP signal is stated to be detected at 14hrs. I would recommend showing a figure of the timelapse instead of just one time point.

- This experiment is carried out with PE-phase bacteria which, as indicated in the manuscript, express the capsule, have you tried using E-phase bacteria?

Figure S6: The shape of the amoebae is concerning and raises the question of their viability. Most of the bacteria seem to be extracellular, this experiment should show intracellular bacteria and a quantification.

- Can you explain why the experiment is done at 37°C while amoebae are usually kept at 20-25°C?

- Could the temperature influence the synthesis of the capsule?

Figure 4 C/D: The experiment in Fig4C shows ΔctrC replication compared to the ΔctrC+empty in FigD. These experiments should be done with both ΔctrC and ΔctrC+empty strains to exclude any effect of the empty plasmid.

- Why different MOI were used for Fig4C/D experiments (figure legend)? Can you comment on why changing the MOI seems to influence the replication of ΔctrC.

- Furthermore, the ΔctrC+compl strain is showing the same behavior as Llo WT while it has been shown that only 50% of the capsule is restored and 60% of the mice are surviving, can you comment?

Overall, the experiments on amoebae need to be strengthened, for example with microscopy images and quantification of intracellular replication to match the paragraph’s title.

Figure 4A/S7A: Is there a particular reason why you used 2 different bacterial loads? Do you have CFUs for the bacterial load of 107 with the ΔctrC+compl?

Figure 4F: The western blot and the flow cytometry data should be provided in a supplemental figure.

Figure 5: The authors should include the complemented strain for these experiments.

Figure S7B: The CFU of t=0 should be provided. I would recommend a later point for the ΔctrC to know if the mice are clearing the infection or if the infection is stabilized over time .

Figure S7C: Related to this figure, it would be interesting to know if the capsule of L. longbeachae protects from complement-mediated binding/killing.

Figure 6: It would have been nice to know the cytokine production in mice.

Reviewer #2: - Why were only female mice used in this study? Can we expect sex differences?

- legend fig.4 “A. castellanii trophozoites were infected at MOI XYZ” – seems like the placeholder xyz needs to be replaced by actual MOI

- the figures are not always very reader-friendly. The fonts can be too small (especially 3A), the graph lines can be very thin (Fig.4) and the panels not sufficiently self-explanatory (e.g. can the panels in Fig.4 be annotated so it’s immediately obvious whether these experiments were carried out in amoeba or mice without having to refer to the figure legend)

- along those lines: could the mouse and amoeba data be shown in two separate figures? I would also recommend moving some of the supplementary data into the main figure. FigS7A-C provides more information than current Fig.4A. I would also recommend showing Fig.S8 as a main figure, because I think it is important to contrast the growth defect of ctrC mutanat in amoeba with the lack thereof in macrophages

- Fig.4E y axis label – percent survival of what? Display competitive index data instead?

- Fig.4E – at 48 hpi WT completely outcompeted the ctrC mutant; the graph only shows 0 and 48 hpi data (all the later timepoints are superfluous and can be deleted) – to show these data as a line graph, timepoints between 0 and 48 hpi - that actually are intermediate datapoints – need to be included

- Fig.4E – this experiment was done as an n of 2?

- fig6 figure legend – I think it should be A-C) hMDMS not A- D)

- Figure 6 – in panels A – C cytokine data is depicted as fold change; in all other panels as mass/ volume (pg/mL). I would recommend showing data as mass/ volume for panels A- C as well, since mas/ volume data are much more informative than fold changes, e.g. by allowing readers to assess whether any of the cytokine measurements are in the physiological range

- ctrC mutant phenotypes were complemented for experiments in in hMDMs (Fig6 A-C) but not in murine DCs (D-F) or murin BMDM (G-I). Ideally complementation should be done for experiments in murine DCs/ BMDMs (D – I) as well

- Fig. 6 why does Llo not induce TNFalpha in THP1 cells when it does so robustly in murine BMDMs? Could the authors provide a positive control demonstrating that their THP1 cells can be activated for TNFalpha production? Without such a control the data shown in Fig6C are difficult to interpret

Reviewer #3: Minor comments

1. Line 242, there is no difference not a slight reduction in replication (Fig. S8)

2. Line 240 …. indicating that complementation partially restored virulence

3. Line 1119 MOI XYZ?

4. Line 1121 percentage survival of amoebae or bacteria?

5. Fig. 4F. What does percent secretion mean? This result is usually expressed as ratio blue/green fluorescence I think

6. Line 1144 …A-C not A-D

PLOS authors have the option to publish the peer review history of their article (what does this mean?). If published, this will include your full peer review and any attached files.

Reviewer #1: **Yes: **Charles Van der Henst & Alexandra Maure

Reviewer #2: No

Reviewer #3: No
---

## [Decision Letter · Decision Letter 1]

26 Aug 2024

Dear Dr. Buchrieser,

We are pleased to inform you that your manuscript 'The unique Legionella longbeachae capsule favors intracellular replication and immune evasion' has been provisionally accepted for publication in PLOS Pathogens.

Best regards,

Tomoko Kubori, Ph.D.

Academic Editor

PLOS Pathogens

Matthew Wolfgang

Section Editor

PLOS Pathogens

Michael Malim

Editor-in-Chief

PLOS Pathogens

orcid.org/0000-0002-7699-2064

The reviewers are fully satisfied with the revised manuscript and have no further concerns.

Reviewer Comments (if any, and for reference):

Reviewer's Responses to Questions

**Part I - Summary**

Reviewer #1: Dear,

I can see that the authors performed additional experiments and that the manuscript has improved significantly.

Reviewer #2: The reviewers addressed my main concerns. The paper makes an important discovery for the Legionella field and I fully support publication of the revised MS in PLoS Pathogens

**Part II – Major Issues: Key Experiments Required for Acceptance**

Reviewer #1: NA

Reviewer #2: (No Response)

**Part III – Minor Issues: Editorial and Data Presentation Modifications**

Reviewer #1: Fig S11: There is no white arrow on the images. Please add them or remove it from the legend.

Reviewer #2: (No Response)

PLOS authors have the option to publish the peer review history of their article (what does this mean?). If published, this will include your full peer review and any attached files.

Reviewer #1: **Yes: **Charles Van der Henst

Reviewer #2: No

---

## [Editor Report · Acceptance letter]

7 Sep 2024

Dear Dr. Buchrieser,

We are delighted to inform you that your manuscript, "The unique *Legionella longbeachae* capsule favors intracellular replication and immune evasion," has been formally accepted for publication in PLOS Pathogens.

Best regards,

Michael Malim

Editor-in-Chief

PLOS Pathogens

orcid.org/0000-0002-7699-2064